# Unidirectional guided-wave-driven metasurfaces for arbitrary wavefront control

Shiqing Li[1], Kosmas L. Tsakmakidis [2] ✉, Tao Jiang[3], Qian Shen[1], Hang Zhang[1], Jinhua Yan[1], Shulin Sun[4] & Linfang Shen [1] ✉

Metasurfaces are capable of fully reshaping the wavefronts of incident beams in desired manners. However, the requirement for external light excitation and the resonant nature of their meta-atoms, make challenging their on-chip integration. Here, we introduce the concept and design of a fresh class of metasurfaces, driven by unidirectional guided waves, capable of arbitrary wavefront control based on the unique dispersion properties of unidirectional guided waves rather than resonant meta-atoms. Upon experimentally demonstrating the feasibility of our designs in the microwave regime, we numerically validate the introduced principle through the design of several microwave meta-devices using metal-air-gyromagnetic unidirectional surface magneto-plasmons, agilely converting unidirectional guided modes into the wavefronts of 3D Bessel beams, focused waves, and controllable vortex beams. We, further, numerically demonstrate sub-diffraction focusing, which is beyond the capability of conventional metasurfaces. Our unfamiliar yet practical designs may enable full, broadband manipulation of electromagnetic waves on deep subwavelength scales.

Light can be confined at the nanoscale through coupling with surface plasmons[1]. The 'two-dimensional' nature of the resulting surface plasmon polaritons (SPPs) offers significant flexibility in engineering photonic integrated circuits for optical communications and optical computing[2–4]. However, the backscattering of SPPs by disorders, defects and structural imperfections, due to the reciprocity of plasmonic platforms, limits their applications in optical systems. Developing nonreciprocal plasmonic platforms[5] that enable unidirectional SPP propagation is, therefore, of great importance. Such unidirectional SPPs occur in nonreciprocal plasmonic platforms made of magnetized semiconductors in the terahertz regime[6] or magnetized metals in the visible regime[7], possessing gyroelectric anisotropy induced by an external magnetic field. These unidirectional SPPs, known as unidirectional surface magnetoplasmons (USMPs) for several decades, have recently regained interest in the context of topological

electromagnetics[7–15]. It has been revealed that true USMPs are topologically protected[12,13] and thus robust against nonlocal effects. These USMPs can be immune to backscattering at disorders due to the absence of a back-propagating mode in the system. In the microwave domain, nonreciprocal materials such as yttrium-iron-garnet (YIG) generally exhibit gyromagnetic anisotropy under an external magnetic field, which can also support unidirectional surface polaritons possessing almost the same guiding properties as USMPs[16–18]. Thus, these low-frequency unidirectional electromagnetic (EM) modes are also referred to as USMPs. Robust USMPs provide a fundamental mechanism for realizing a plethora of optical devices that are impossible using conventional reciprocal EM modes, such as optical cavities that overcome the time-bandwidth limit[19].

Relying on the bandgap of nonreciprocal materials themselves, USMPs easily attain broad bandwidth and simultaneously exhibit

[1]Department of Applied Physics, Zhejiang University of Technology, Hangzhou 310023, China. [2]Section of Condensed Matter Physics, Department of Physics, National and Kapodistrian University of Athens Panepistimioupolis, Athens GR-157 84, Greece. [3]Yangtze Delta Region Institute (Huzhou), University of Electronic Science and Technology of China, Huzhou 313001, China. [4]Shanghai Engineering Research Centre of Ultra Precision Optical Manufacturing, Department of Optical Science and Engineering, School of Information Science and Technology, Fudan University, Shanghai 200433, China. ✉e-mail: ktsakmakidis@phys.uoa.gr; lfshen@zjut.edu.cn

unique dispersion properties. For example, in the microwave range, the dispersion curve of USMPs in YIG materials can monotonically grow across the entire light cone in air. This implies that USMPs can smoothly convert from waves with positive velocities to waves with negative velocities without any frequency gap while they maintain the sign of their group velocities. As the dispersion of USMPs is closely related to the structural details and, hence, can be flexibly tailored through controlling the structural parameters. For USMP at a fixed frequency, the phase constant can be tuned over the range $[-k_0, k_0]$ (where $k_0$ is the free-space wavenumber). Based on this unparalleled phase controllability, it is interesting to investigate whether USMPs may offer a unique mechanism for designing metasurfaces, one that would have no resonant nature. Metasurfaces[20–22] are capable of reshaping the wavefronts of incident beams in desired manners, and they are thought to be great candidates for revolutionizing conventional optics. Relying on engineered optical resonators, known as meta-atoms, metasurfaces can locally provide abrupt phase shifts at subwavelength intervals to tailor the phase of incident waves, making them exhibit predetermined functions for transmitted or reflected waves based on the Huygens principle. These metasurfaces can provide unparalleled control over EM waves to realize complex free-space functions, such as beam deflection[23,24], focusing[25–27], generation of orbital angular momentum beams[28], and holograms[29]. However, most metasurfaces are driven by free-space waves, making their on-chip integration challenging. Moreover, based on the resonant mechanism of the meta-atoms, the designed wavelengths of metasurfaces are closely related to the sizes of the meta-atoms, making it difficult to shape wavefronts on deep subwavelength scales. These metasurfaces also generally operate within a narrow frequency range.

In this work, we develop a fresh class of metasurfaces that are driven by USMPs (Fig. 1) based on their unparalleled phase con-

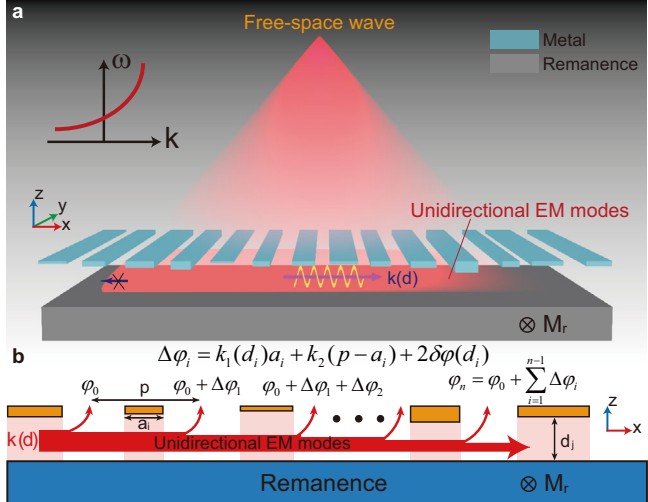

**Fig. 1 | Working principle of unidirectional guided-wave-driven metasurfaces. a** Schematic of a unidirectional guided-wave-driven metasurface. The phase and intensity of the extracted wave from unidirectional waveguide by each meta-cell can be tuned individually. An array of meta-cells work collaboratively to form certain wavefronts and fulfill different functions, such as beam focusing. **b** Illustration of the wavefront formation of the extracted wave. The phase difference of the extracted waves from two neighboring cells is contributed from two parts: the phase accumulation $k_1(d_i)a_i + k_2(p-a_i)$ from the unidirectional guided-wave propagation and the phase shift $2\delta\varphi(d_i)$ from the coupling of two unidirectional guided modes with each other in each cell. As a result, the phase difference of the extracted waves from two neighboring cells can be expressed as $\Delta\varphi_i = k_1(d_i)a_i + k_2(p-a_i) + 2\delta\varphi(d_i)$. The superposition of phase difference $\Delta\varphi_i$ constitute the phase of extracted wave from the $n$-th cell $\varphi_n = \varphi_0 + \sum_{i=1}^{n-1}\Delta\varphi_i$, where $\varphi_0$ is the initial phase of the incidence.

trollability. The constituent meta-cells of such USMP-driven meta-surfaces support two types of USMPs, and during their conversion, directional radiation is created, extracting guided waves into free space and shaping them into desired light fields. Based on the robustness of USMPs, the proposed metasurfaces would be utterly immune to backscattering from surface-roughness. In contrast to existing guide wave-driven metasurfaces[30–32], our USMP-driven metasurfaces provide desired spatial phase profiles for extracted waves directly from the propagation of USMPs, thus do not need any meta-atoms to induce abrupt phase shifts. Although plasmonic metasurfaces based on Bragg interferences also work without meta-atoms[33–38], they can only manipulate wavefronts at wavelength scales due to the limited phase controllability of conventional SPPs. Very recently, leaky-wave metasurfaces (LWM) have been proposed[39,40], which are constructed by directly opening holes on conventional metal waveguides. It controls phase and amplitude of extracted wavefront through hybridizing local leaky waves staggered by a quarter of guided wavelength within the constituent meta-cell. However, such metasurfaces can only manipulate wavefronts at wavelength scales. Our metasurfaces represent a fresh class of LWMs distinguished by a unique mechanism. Leveraging the singular dispersion and robust unidirectionality of USMPs, these metasurfaces possess the capability to flexibly modulate both the phase and amplitude of leaky waves by fine-tuning local waveguide parameters. This capability enables precise manipulation of wavefronts, even at deep subwavelength scales. The developed technology opens up exciting possibilities for constructing multifunctional USMP-driven meta-devices with flexible access to free space, providing advantages such as ease of fabrication, reconfiguration, and compatibility with on-chip technology. Our magneto-optical structures are also amen-able to a plethora of further effects allowing potentially for high-speed modulation[41–45]. This technology has the potential to stimulate numerous related applications, including communications, remote sensing, and virtual reality displays.

## Results
### Phase controllability of USMPs

We first investigate nonreciprocal waveguide systems that can support USMPs. The material configuration plays a critical role in determining the properties of USMPs, and we consider two types of nonreciprocal waveguides, as illustrated in Fig. 2a: a YIG-dielectric-metal layered structure (type-I waveguide) and a YIG-air layered structure (type-II waveguide). Both types of waveguides can support USMPs at microwave frequencies. For the type-I waveguide, we assume that the dielectric has a relative permittivity close to 1 (e.g., foam). In both waveguides, the YIG ($\varepsilon_m = 15$)[46] possesses remanent magnetization $\vec{M}_m = \vec{y}M_m$ that breaks the time-reversal symmetry of the system, enabling the existence of USMP in the absence of external magnetic field. The remanence induces gyromagnetic anisotropy in the YIG, with the permeability tensor taking the form

$$\mu_m = \begin{bmatrix} \mu_1 & 0 & -i\mu_2 \\ 0 & 1 & 0 \\ i\mu_2 & 0 & \mu_1 \end{bmatrix}, \tag{1}$$

where $\mu_1 = 1$, and $\mu_2 = \omega_m/\omega$ ($\omega$ is the angular frequency). Here, $\omega_m = 2\pi f_m = \mu_0 \gamma M_m$ is the characteristic circular frequency ($\mu_0$ is the vacuum permeability, $\gamma$ is the gyromagnetic ratio). Surface magneto-plasmons (SMPs) in the type-I waveguide have a dispersion property closely depending on the dielectric-layer thickness ($d$), which is described by the dispersion relation

$$\alpha_r\mu_v + \left(\alpha_m + \frac{\mu_2}{\mu_1}k\right)\tanh(\alpha_r d) = 0, \tag{2}$$

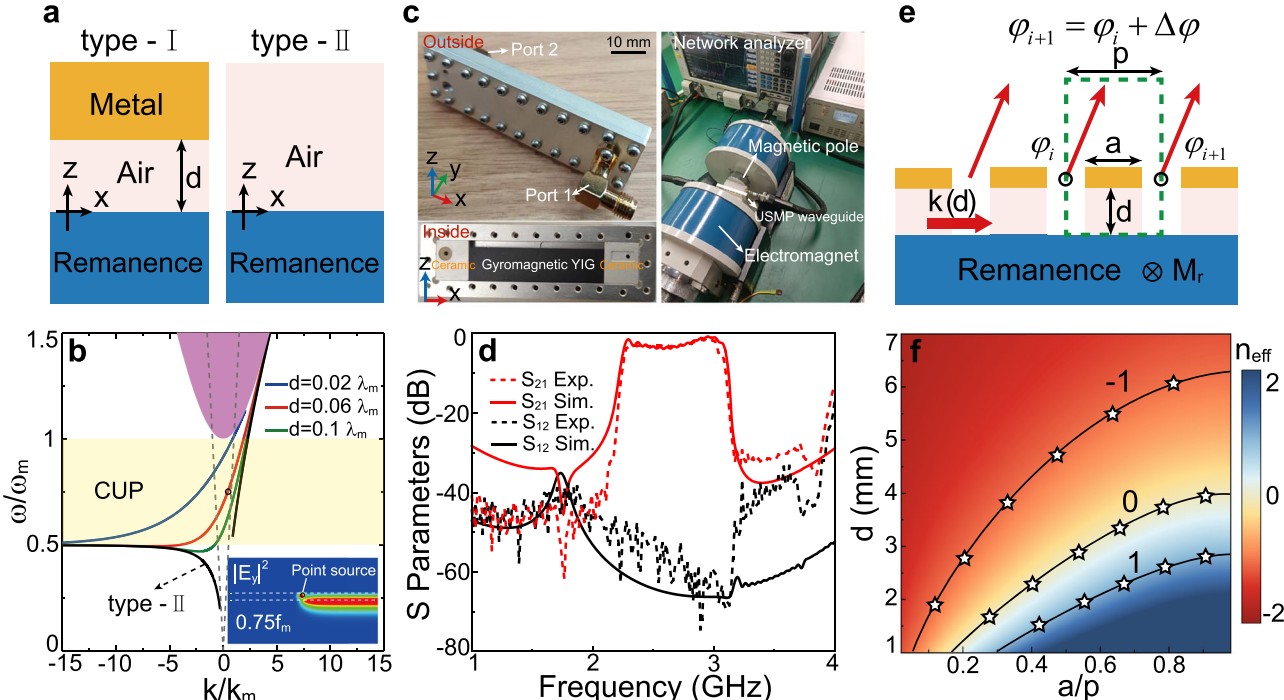

**Fig. 2 | Phase controllability of the unidirectional guided wave. a** Schematic diagram of waveguides supporting type-I and type-II USMPs at microwave frequencies, and (**b**) the dispersion relations of USMPs for various thicknesses of the dielectric layer. The shaded rectangular area indicates the unidirectional frequency window for the waveguide, and the other shaded areas indicate the zones of bulk modes in the gyromagnetic materials ($\varepsilon_m = 15$). Inset: simulated electric-field intensity distribution excited by a line current source placed in the middle of waveguide structure with $d = 0.06\lambda_m$ and frequency $\omega = 0.75\omega_m$. **c** Left: photograph of the fabricated waveguide sample. Right: photograph of the measurement configuration. **d** Simulated (solid) and measured (dashed) S21 and S12 parameters of

the waveguide sample displayed in (**c**). **e** Schematic picture of the periodically uniform metasurface formed by alternating type-I and type-II waveguides shown in (**a**). The period length, duty cycle, and dielectric-layer thickness are denoted by $p$, $a/p$, and $d$, respectively. The phases of extracted waves from two neighboring meta-cells are $\varphi_i$ and $\varphi_{i+1}$ with a difference $\Delta\varphi$. **f** Pseudocolor map of simulated $n_{\text{eff}} = \Delta\varphi_{\text{cl}}/(k_0 p)$ for the structure of (**e**) in a parameter space spanned by dielectric thickness ($d$) and duty cycle ($a/p$) at 2.69 GHz. The three black lines indicate $n_{\text{eff}}$ covering [−1, 1] range. The black stars indicate the theoretical results obtained from Eq. (4).

where $k$ is the phase constant, $\alpha_m = \sqrt{k^2 - \varepsilon_m \mu_v k_0^2}$ with $\mu_v = \mu_1 - \mu_2^2/\mu_1$, $\alpha_r = \sqrt{k^2 - k_0^2}$ for $|k| > k_0$ and $\alpha_r = -i\sqrt{k_0^2 - k^2}$ for $|k| \leq k_0$. The dispersion relation for the type-II waveguide can be directly obtained from Eq. (2) by letting $d = \infty$, which gives

$$\alpha_r \mu_v + \alpha_m + \frac{\mu_2}{\mu_1}k = 0 \qquad (3)$$

The linear terms with respect to $k$ in Eqs. (2), (3) imply that the two waveguides are intrinsically non-reciprocal, allowing for the possibility of unidirectional wave proapgation.

The dispersion Eq. (2) for the type-I waveguide was numerically solved, yielding results displayed in Fig. 2b, where various values of $d$ were analyzed. The dispersion relation for SMPs in the type-I waveguide exhibits a single branch that extends across the entire light cone in air. The positive slop of the dispersion curve unambiguously establishes the USMP (type-I USMP) nature of SMPs. Moreover, the dispersion curve descends as $d$ increases, allowing for control of the phase constant $k$ by adjusting the thickness of the dielectric layer. In all cases, type-I USMPs possess an asymptotic frequency of $\omega_{\text{smp}} = 0.5\omega_m$, which sets a lower-frequency cutoff. However, robust unidirectional propagation is achieved exclusively in the ferrite bandgap, where no backward-propagating mode exists. Consequently, robust type-I USMP occurs within the frequency range $[0.5\omega_m, \omega_m]$. The dispersion relation (3) for the type-II waveguide is also depicted in Fig. 2b, showing two asymmetric

branches: one with $k < 0$ and the other with $k > 0$. Both dispersion branches lie beyond the light cone in air. The branch with $k < 0$ has an asymptotic frequency of $\omega_{\text{smp}} = 0.5\omega_m$, setting its upper-frequency cutoff. As a result, USMP (type-II USMP) occurs in the range of $\omega_{\text{smp}} > 0.5\omega_m$. The type-II USMP lacks robustness when encountering obstacles or disorder, resulting in partial scattering into free space. Clearly, within the range of our concern $[0.5\omega_m, \omega_m]$, both types of waveguides support USMPs. In contrast to type-II USMP with $k > k_0$, type-I USMP possesses a tunable phase constant $k$ range over $[-k_0, k_0]$ by varying $d$. For instance, at $f = 0.75f_m$, $k$ increases from $-2.93k_0$ to $1.9k_0$ as $d$ grows from $0.02\lambda_m - 0.2\lambda_m$.

Then, we further realize the nonreciprocal waveguide in experiment (see Methods for details). The photographs of the fabricated nonreciprocal waveguide and the schematic of the measurement configuration are shown in Fig. 2c, here, magnetized YIG serves as the gyrotropic medium. The measured frequency-dependent S parameters of the prototype of the waveguide compared with the simulated S parameters are shown in Fig. 2d. The experimental results clearly show that there is a wide band of unidirectional propagation in the frequency range from 2.3 – 3.1 GHz (fractional bandwidth about 30%) with reverse isolation of 40 dB. Within the unidirectional waveguide, the propagation loss of USMP is mainly caused by the loss of the YIG material. At the center (2.7 GHz) of the unidirectional window, the measured propagation length (over which the energy flow decays by 1/$e$) of USMP is about $8\lambda_0$ (vacuum wavelength). The unique dispersion of the unidirectional mode was also experimentally demonstrated, and it can be adjusted by varying the external magnetic field (see Supplementary Fig. S1). The measured results agree well with the simulated

results. Despite the experimental limitations that restricted our ability to demonstrate more complex effects, the verified unidirectional propagation property, small propagation loss, and complete phase controllability serve as the foundation of our proposed unidirectional guided-wave-driven metasurfaces, providing compelling evidence for the feasibility of the proposed concept.

Next, we investigate a periodic structure that consisting of alternating type-I waveguide of length $a$ and type-II waveguide of length $b$, as depicted in Fig. 2e. The unit cell of this structure, with a subwavelength length of $p = a + b$, supports both types of USMPs and is also referred to as meta-cell. Evidently, such structures with deep-subwavelength $d$ ($\sim \lambda/30$) can be utilized as metasurfaces for beam deflection, which extracts waves into free space by scattering two types of USMPs at their interfaces. The extracted wave has uniformly distributed phases ($\varphi_i$) spaced by $\Delta\varphi_{cl}$, which equals to the accumulated phase of USMPs traveling over a unit cell, given by

$$\Delta\varphi_{cl} = k_1 a + k_2 b + 2\delta\varphi_{12},\tag{4}$$

where $k_1$ is the phase constant of type-I USMP that is determined by $d$, and $k_2$ is the phase constant of type-II USMP. $\delta\varphi_{12}$ represents the phase shift from the coupling of two USMPs, which is far smaller than $\pi$ in general (see Supplementary Fig. S2). The subwavelength period of the metasurface eliminates high-order diffractions, resulting in a well-defined angle of extracted beam given by $\theta = \arcsin(k_x/k_0)$, where $k_x = \Delta\varphi_{cl}/p$, also known as the Bloch wavevector. Obviously, the output angle $\theta$ can be effectively adjusted by the parameter $d$, offering a beam deflection angle range of nearly 180° based on the phase controllability of type-I USMP (see Supplementary Fig. S3). On the other hand, noting that $\Delta\varphi_{cl}$ can also be tuned by adjusting parameter $a$. The $\Delta\varphi_{cl}$ was numerically calculated using full-wave finite element method (FEM), and the results are shown in Fig. 2f for $f_m = 3.587$ GHz, $f = 0.75 f_m$ and $p = 36$ mm. Here, to characterize the phase controllability of such metasurfaces, we introduce an effective index that is $\Delta\varphi_{cl}$ scaled by $k_0 p$,

$$n_{eff} = \Delta\varphi_{cl}/(k_0 p)\tag{5}$$

As displayed in Fig. 2f, this effective index can be tuned within a range extending beyond [−1, 1], validating the phase controllability of the metasurfaces.

Let's suppose an incident wavefront in free space at $z = 0$, traveling along the positive $z$ direction. We represent the complex field across the wavefront using $U(x, 0)$ and express it as a Fourier integral

$$U(x,0) = \int_{-\infty}^{+\infty} A(k_x) \exp(jk_x x)dk_x,\tag{6}$$

where $A(k_x)$ is the spatial spectrum of the field. The consequent field $U(x, z)$ in free space ($z > 0$) can be expressed in terms of the spatial spectrum, yielding

$$U(x,z) = \int_{-k_0}^{k_0} A(k_x)e^{i(k_x x + k_z z)}dk_x + \left[\int_{-\infty}^{-k_0} + \int_{k_0}^{+\infty}\right]A(k_x)e^{(jk_x x - \gamma z)}dk_x,\tag{7}$$

where $k_z = \sqrt{k_0^2 - k_x^2}$ for $|k_x| \leq k_0$ and $\gamma = \sqrt{k_x^2 - k_0^2}$ for $|k_x| > k_0$. Evidently, the field $U(x, z)$ can be divided into two parts: propagating wave with $|k_x| \leq k_0$ and evanescent wave with $|k_x| > k_0$, which are represented by the first and second terms in Eq. (7), respectively. The evanescent part attenuates rapidly along $z$, and therefore, the field pattern in far field is mainly determined by the propagating part. Thus, for achieving any desired free-space mode, we only need to construct initial wavefront $U_0(x)$ with a spatial spectrum lying within the region $|k_x| \leq k_0$, i.e.,

$$U_0(x) = \int_{-k_0}^{k_0} A(k_x) \exp(jk_x x)dk_x\tag{8}$$

For such a wavefront, the phase difference of the field between any two points separated by $p$ should be less than $k_0 p$, i.e., $-k_0 p \leq \arg[U_0(x+p)] - \arg[U_0(x)] \leq k_0 p$ (further discussed in Supplementary Fig. S4). For our USMP-driven metasurface, the difference of adjacent extracted phases separated by the subwavelength distance $p$ can be tuned over the range of $[-k_0 p, k_0 p]$, which corresponds to the $n_{eff}$ tune range of meta-cells larger than the interval [−1, 1], so it would be suffice to implement free-space propagating wave modulation. This also greatly distinguishes from the existing metasurfaces which necessitate phase shifts cover the entire $2\pi$ phase range to achieve complete control of wavefront. In the USMP-driven metasurfaces, the scattering strength of USMPs depends on the mismatch between the modal profiles of the two USMPs. The modal spot of type-I USMP is closely related to $d$, allowing for scattering amplitude adjustment by controlling $d$. Additionally, the amplitude of the extracted wave can be significantly impacted by the size of the radiation aperture, characterized by the length $b$. As shown in Fig. 2f, for any desired phase ($n_{eff}$), $b$ can be chosen from a wide range [0, 0.7$p$], providing ample operational freedom for amplitudes. Hence, by selecting appropriate $d$ and $a$ values, we can simultaneously achieve desired extracted phases and amplitudes. By constructing nonuniform periodic structures with meta-cells of the same length $p$ but different $d$ and $a$ values, we can achieve spatial-variant optical responses that extract and mold USMPs to any desired free-space optical modes.

To demonstrate the capabilities of USMP-driven metasurfaces, we conducted numerical simulations showcasing wave focusing and Bessel-beam generation directly from USMPs in waveguide. For simplicity, we utilized YIG material with remanence to construct metasurface cells capable of supporting USMPs without the need of an external magnetic field. Leveraging the unique characteristics of USMP-driven metasurfaces, we also presented numerical evidence of sub-diffraction focusing, a feat unattainable with conventional metasurfaces. Additionally, by using meta-cells to construct a ring cavity, we can generate vector optical vortices with quantized orbital angular momentums (OAMs). Furthermore, the OAM order of the radiated vortex beam can be tuned by varying the frequency or even by tailoring the external magnetic field at a fixed frequency.

## Wave focusing and Bessel-beam generation

In order to construct unidirectional guided wave-driven metasurfaces capable of complex 3D manipulation of free-space waves, 3D meta-cells are required as building blocks. By terminating 2D meta-cells in the $y$ direction with a pair of subwavelength-separated metal slabs (which can be approximated as perfect electric conductors in the microwave regime), the resulting 3D meta-cells are nearly physically identical to their 2D counterparts. Leveraging these designed 3D meta-cells, it becomes possible to construct unidirectional guided wave-driven metasurfaces with desired free-space functions, as illustrated in Fig. 3a. Along the $x$ axis, 3D USMP-driven metasurface is physically identical to its 2D counterpart, i.e., arbitrary $\Delta\varphi(x, y)$ along the $x$ axis can be achieved through tailoring the height of the upper metal slabs $d(i, j)$ ($i, j = 1, 2, 3, ...$) of 3D meta-cells. To realize arbitrary 3D modulation of extracted wave, it is also necessary to have full control over the initial phase along the $y$ axis, denoted as $\varphi(0, j)$. Evidently, through tailoring the heights $d(0, j)$ of the upper metal slabs of the input (type-I) waveguides to achieve different phase accumulations from the propagation of USMPs, the arbitrary initial phase $\varphi(0, j)$ of extracted wave can be achieved. At any rate, leveraging the unique dispersion property of type-I USMPs, 3D meta-cells can offer excellent phase

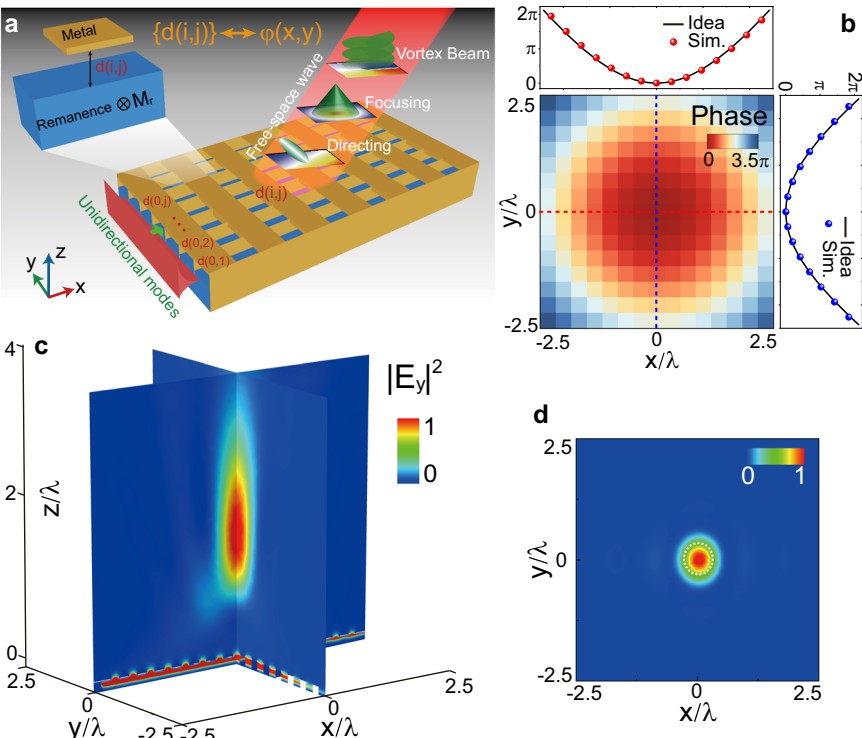

**Fig. 3 | Demonstration of 3D manipulation of free-space wave with a unidirectional guided wave-driven metasurface. a** Schematic of the designed metadevice which is composed of 2D meta-cells terminated in the $y$ direction with a pair of metal slabs separated by a subwavelength distance. Through modulate the height distribution of the upper metal slabs $d\{i, j\}$, extracted wave from the unidirectional waveguide can fulfill arbitrary functions, such as free-space beam directing, focusing and vortex beam generation. **b** The simulated phase distribution of extracted wave from each meta-cell of the designed meta-device, together with its corresponding horizontal and vertical cuts. The target phase profiles along both the horizontal and vertical directions are also presented for comparison. **c** Simulated 3D electric field intensity $|E_y|^2$ distribution of the designed meta-device, the extracted wave converged at designed focal point (2.24 $\lambda$ above the meta-device) at 2.69 GHz. **d** The simulated $|E_y|^2$ distributions on the $xy$ plane with $z = 2.24 \lambda$, with the dashed-line circle defining the size of the focal spot. All field values are normalized against the maximum value in the corresponding pattern.

controllability through modulation of the height distribution of the upper metal slabs $d\{i, j\}$. This enables the extracted wave to fulfill arbitrary free-space functions, such as 3D beam directing, focusing, and vortex beam generation, as illustrated in Fig. 3a.

Let's take 3D beam focusing as an example. Through designing a USMP-driven metasurface to fulfill a 3D lens phase function $\varphi(x,y) = -k_0 \left( \sqrt{x^2 + y^2 + F^2} - F \right)$, we can focus the extracted wave in free space with a designated focal length $F$. According to Eq. (4), the phase shift provided by a meta-cell between the coordinates $(x - p)$ and $x$ should be

$$\Delta \varphi_{cl} = -k_0 \left[ \sqrt{x^2 + y^2 + F^2} - \sqrt{(x - p)^2 + y^2 + F^2} \right] \quad (9)$$

Evidently, based on $\Delta \varphi_{cl}$, structure parameters $d\{i, j\}$ can be determined by searching the database of Fig. 2d ($a/p = 0.5$ here), and the meta-device can be achieved by spatially arranging 3D meta-cells with the designed $d\{i, j\}$. We simulated such a metalens using a fullwave FEM at 2.69 GHz. Figure 3b shows the phase distribution $\varphi(x, y)$ of extracted wave for realizing a 3D focusing with $F = 250$ mm ($\sim 2.24\lambda$) and $p = 36$ mm, the duty cycle $a/p$ of all meta-cells is fixed as 0.5. Evidently, the phase of the extracted wave satisfies a parabolic distribution. Furthermore, the simulated phase of extracted electric field agrees well with that from our theoretical calculation, along not only the horizontal direction (Fig. 3b, top), but also the vertical direction (Fig. 3b, right), further validating our USMP-driven metasurface approach. The simulated $|E_y|^2$ field distributions in both $xz$ and $yz$ planes are depicted in Fig. 3c. Obviously, unidirectional guided-wave is

extracted and focused into free space by the metalens, with a focal length $F = 2.23 \lambda$ in good agreement with theoretical prediction 2.24 $\lambda$. To check the quality of the focusing effect, we quantitatively evaluate the full width at half maximum (FWHM) of the focal spot on the focal plane, and find it is approximately 0.59 $\lambda$ (Fig. 3d). Such a value strongly depends on the aperture size of our metalens, and can be further reduced by enlarging the total size of our metalens. Evidently, by simply changing the distribution of $d\{i, j\}$, the focusing function of this metadevice can be switched to other functions such as beam directing and vortex beam generation.

Bessel beams, which are solutions of the free-space Helmholtz equation, exhibit unique transverse amplitude distributions that can be described by the Bessel functions of the first kind. These beams possess remarkable properties, including non-diffraction and self-reconstruction, and can feature an extremely narrow central spot radius, on the order of one wavelength. These properties make Bessel beams a promising tool for a wide range of applications in optics and photonics. Ideal Bessel beams carry infinite energy and are not spatially limited, their central intense region is surrounded by concentric rings that each contain an energy flux equal to that of any other. As optical systems inherently possess a finite aperture, Bessel beams can only be approximated in a limited region through the superposition of multiple plane waves. This approximation can be achieved using axicons, such as conical prisms, which refract light rays symmetrically toward the optical axis. Alternatively, metasurfaces have been proposed as a means to generate Bessel beams, with these structures being referred to as meta-axicons. However, most of these meta-axicons are driven by waves in free space, which poses challenges for on-chip integration. Here, we demonstrate the generation of Bessel

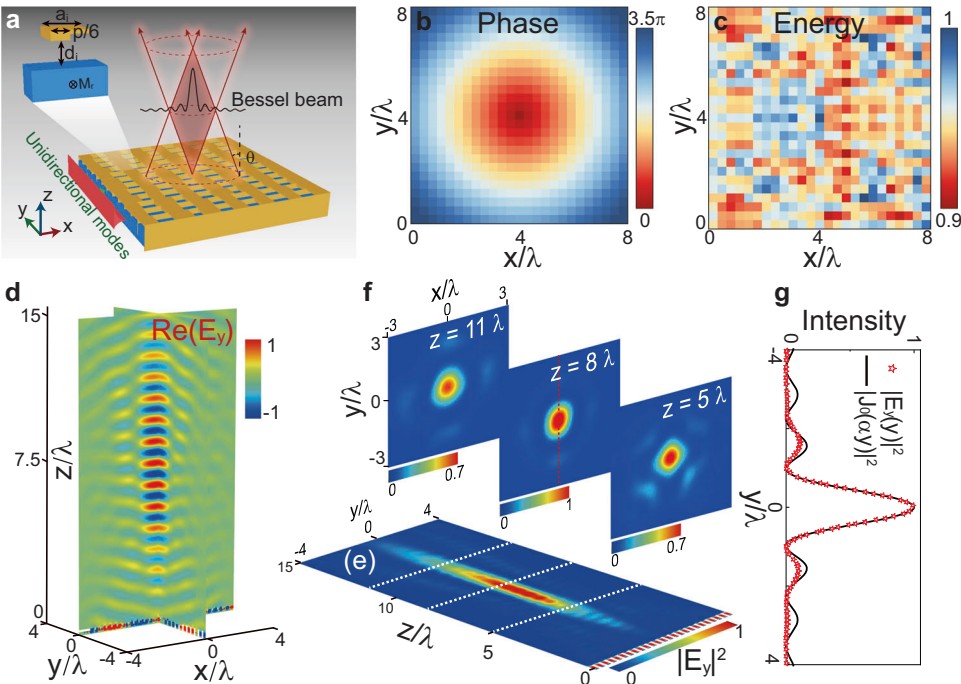

**Fig. 4 | Demonstration of free-space Bessel-beam generation with a unidirectional guided-wave-driven metasurface. a** Schematic diagram of the USMP-driven meta-axicon which is composed of L-shaped meta-cells. The scattering wave from unidirectional waveguide is deflected to an angle $\theta$ toward its center to generate Bessel beam. **b, c** The simulated phase and energy of the extracted wave from each meta-cell of the designed meta-axicon. **d** Simulated 3D $Re(E_y)$ distribution for our meta-device (placed at $z = 0$ mm) driven by unidirectional guided-wave at 2.69 GHz. **e** Normalized intensity profiles $|E_y|^2$ of the generated Bessel beam in $yz$ plane with $x = 0$ and (**f**) $xy$ planes with $z = 5\lambda$, $8\lambda$, and $11\lambda$. **g** Normalized $|E_y|^2$ distributions along the line with $z = 8\lambda$ and $x = 0$, and the comparison with theoretical formula for a zero-order Bessel functions $J_0(\alpha y)$.

beams directly from photonic integrated components using USMP-driven metasurfaces.

As a proof of concept, we numerically demonstrate the generation of zero-order Bessel beam using a USMP-driven meta-axicon (Fig. 4a). Such a meta-axicon requires a spatial phase profile of the form $\varphi(x,y) = -\alpha\sqrt{x^2 + y^2}$, and the phase shift provided by a meta-cell between the coordinates $(x - p)$ and $x$ should be

$$\Delta\varphi_{cl} = -\alpha\left[\sqrt{x^2 + y^2} - \sqrt{(x - p)^2 + y^2}\right], \qquad (10)$$

where $\alpha = k_0\sin\theta$ is the transverse wave number, and $\theta$ is the angle at which radiation rays cross the optical axis. To flexibly control both the phase and amplitude of extracted waves, we employ L-shaped meta-cells with $p = 36$ mm to construct the meta-axicon. The design process of this USMP-driven meta-axicon is detailed in the Method section. The Bessel beam generated by such meta-axicon is numerically simulated using full-wave FEM. Figure 4b and c respectively display the simulated phase and energy of the extracted wave at the designed frequency $f = 2.69$ GHz. Evidently, simulated electric field distribution for the designed meta-axicon validates that the extracted waves have almost uniform amplitudes and spatial-variant phases as described by $\varphi(x,y) = -\alpha\sqrt{x^2 + y^2}$.

Figure 4d shows the simulated $Re[E_y]$ distributions at both $xz$ and $yz$ planes for the transmission of extracted wave in free space, and Fig. 4e shows the simulated intensity profile $|E_y|^2$ in $yz$ plane (with $x = 0$). Both Fig. 4d and e clearly show that the extracted wave in this configuration is indeed a well-behaved Bessel beam exhibiting a clear nondiffracting feature. The propagation range of the Bessel beam is observed to be $y_{max} = 1204.6$ mm. This maximum propagation length is close to the theoretical value using geometric optics, that is $L/(2\tan\theta) = 1236.4$ mm, where $L = 900$ mm is the length of the meta-

axicon region. To characterize the performance of the generated Bessel beam, we also displayed the intensity distribution $|E_y|^2$ of the extracted wave in three $xy$ planes at different longitudinal positions ($z = 5\lambda$, $8\lambda$, and $11\lambda$). As shown in Fig. 4f, the generated transverse field patterns exhibit nice rotationally invariant symmetries with strengths that decay quickly away from the center. In Fig. 4g, we compare the intensity distribution along the $y$ axis (at $z = 8\lambda$ and $x = 0$ mm), obtained by the FEM simulation, with the zero-order Bessel function $J_0(\alpha y)$, where $\alpha = k_0\sin\theta$. Excellent agreement among these results clearly demonstrate the high quality of the Bessel beam generated by our metadevice. The full width at half maximun (FWHM) of this Bessel beam is 110.84 mm, which is close to its theoretical value, given by $\omega_{FWHM} = 2.25/\alpha = 116.76$ mm.

In addition, it is important to evaluate the efficiency of the proposed metasurfaces. In the simulation, the absorption losses of remanence materials are characterized by the damping coefficient $\nu = 10^{-3}\omega$[46]. The simulated utilization efficiencies (UEs) of proposed metasurfaces are 70% for wave focusing and 13% for Bessel-beam generation, which is calculated as the ratio between phase-modulated output and the energy decrease of USMPs inside the waveguide. It should be noted that the lower UEs observed in Bessel beam excitation can be attributed to the relatively larger thickness of the air layer within the latter type-I waveguide, resulting in a smaller amount of extracted energy. In fact, the UEs of Bessel-beam generation can also exceed 70% after parameter optimization (further discussed in Supplementary Fig. S5). The reduced efficiency primarily stems from material absorption losses.

## Sub-diffraction focusing
By utilizing USMP-driven metasurfaces composed of deep-subwavelength meta-cells, we have successfully constructed a metasurface capable of manipulating both propagating and evanescent

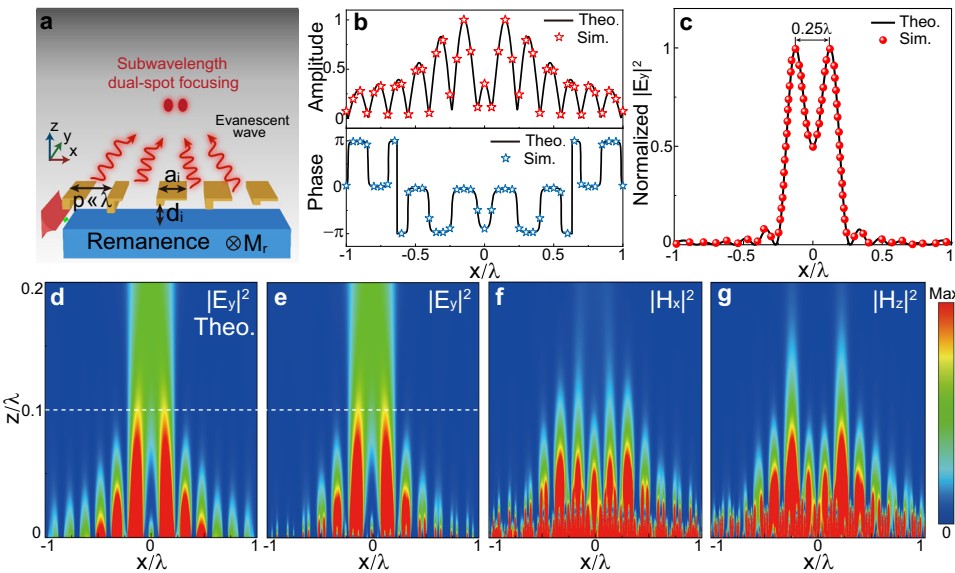

**Fig. 5 | Demonstration of sub-diffraction dual-spot focusing with a USMP-driven metasurface. a** Schematic diagram of the USMP-driven metalens which is composed of 2D L-shaped meta-cells with a deep subwavelength size $p \ll \lambda$. The wavefront extracted from the USMP waveguide can not only contain spatial low-frequency components but also spatial high-frequency components, and the sub-diffraction dual-spot focusing can be realized with the contribution of evanescent fields. **b** Designed amplitude, phase profiles localized, and the simulated amplitude and phase of the extracted wave from each meta-cell. **c** Normalized $|E_y|^2$ at the height $z = 0.1\lambda$, along with the corresponding theoretical results. The two spots are separated by $0.25\lambda$ with each hotspot size of $0.127\lambda$. **d** Theoretically computed field pattern $|E_y|^2$ excited by an initial wavefront $E_y = U_0(x)$ of finite size $2\lambda$ at $z = 0$. **e**–**g** Normalized intensity profiles $|E_y|^2$, $|H_x|^2$ and $|H_z|^2$ of the generated sub-diffraction dual-spot focusing beam.

wave components, thereby achieving sub-wavelength focusing in the near-field regime and ultimately breaking the diffraction limit. Sub-wavelength focusing holds great potential for improving the resolution of imaging systems, enhancing detector sensitivity, and expanding a multitude of applications, including but not limited to superresolution microscopy, nanoscopy, nanolithography, and optical trapping. The key to achieving subwavelength resolution and overcoming the diffraction limit lies in the precise tailoring of the evanescent spectrum of an aperture field distribution. Various methods for achieving super-focusing or super-resolution imaging have been proposed[2,47–55], including superlenses formed with negative refractive index materials[56]. It is essential to acknowledge that all known methods for super-focusing or super-resolution imaging have their specific limitations and drawbacks. Thus, it is crucial to explore additional possibilities that may offer complementary advantages and further enhance our understanding of these phenomena.

It is worth noting that a USMP metasurface can construct an initial wavefront that can not only contain propagating field components but also evanescent field components. Furthermore, it is completely feasible to reduce the meta-cell sizes to a deep-subwavelength level. With these unique characteristics, the USMP-driven metasurface presents a promising approach to realizing near-field subwavelength focusing that cannot be achieved with conventional metasurfaces. To vividly illustrate this, we numerically demonstrate the generation of near-field sub-diffraction dual-spot focusing using a USMP-driven metadevice composed of L-shaped meta-cells (Fig. 5a). Such a metadevice requires an initial wavefront $U_0(x)$ of the form

$$U_0(x) = \int_{-k_0}^{k_0} 2\cos\left(k\frac{\Delta}{2}\right)\cos(kx)e^{-if_0\sqrt{k_0^2-k^2}}dk$$
$$+ \left[\int_{-k_{max}}^{-k_0} + \int_{k_0}^{k_{max}}\right] 2\cos\left(k\frac{\Delta}{2}\right)\cos(kx)e^{f_0\sqrt{k^2-k_0^2}}dk, \tag{11}$$

where $\Delta$ represents the distance between two focal spots, $k_{max}$ determines the hotspot size $d_{FWHM} = 2.783/k_{max}$. The case of $k_{max} = k_0$ corresponds to the usual diffraction-limited focusing without the

contribution of evanescent modes. To achieve sub-diffraction focusing, $k_{max}$ must be greater than $k_0$. We consider $k_{max} = 3.5k_0$ here, which corresponds to a hotspot size of $d_{FWHM} = 0.127\lambda_0$, and the black lines in Fig. 5b illustrate the spatial amplitude and phase profiles described by Eq. (11). The large amplitude and phase fluctuations within a single wavelength range pose a formidable challenge for conventional metasurfaces in attaining near-field subwavelength focusing. This predicament is primarily attributed to the resonant nature of meta-atoms, which makes it difficult to manipulate wavefronts at deep subwavelength scales.

In our design, we adopt L-shaped meta-cells with the deep sub-wavelength size $p = \lambda/20$ to construct the metadevice. The design scheme is similar with the aforementioned meta-axicon. Figure 5e shows the electric field intensity distribution $|E_y|^2$ of the extracted wave from the designed metadevice. Evidently, subwavelength focusing is achieved due to the excitation of evanescent modes. The center-to-center distance of the two spots is only 0.25 wavelength, but the shape of the two spots remains very clear and distinguishable. In Fig. 5c we plot the $|E_y|^2$ distribution along a line crossing the focal point (i.e., at $z = 0.1\lambda$), and the FWHM is calculated as $0.127\lambda$, much smaller than the case when only propagation modes are contributing ($0.443\lambda$), thereby overcoming the diffraction limit. The simulated electric-field distribution on the designed metadevice is displayed in Fig. 5b, further validating that the extracted waves have targeted spatial amplitude and phase profiles described by Eq. (11) (black lines in Fig. 5b). For comparative purposes, the theoretically computed field pattern ($|E_y|^2$) excited by an incident wavefront $E_y = U_0(x)$, as described by Eq. (11), is also plotted in Fig. 5d. Evidently, the simulated electric-field pattern $|E_y|^2$ radiated from the USMP-driven metadevice in Fig. 5e agrees well with the targeted one of Fig. 5d. Evidently, given such (deep-sub-wavelength) manipulation capability, one can design holographic metasurfaces capable of generating images with sub-diffraction resolution in the near field. An exemplary demonstration was provided via numerical simulation, showcasing super-resolution imaging of the Greek letter "φ" (see Supplementary Fig. S6).

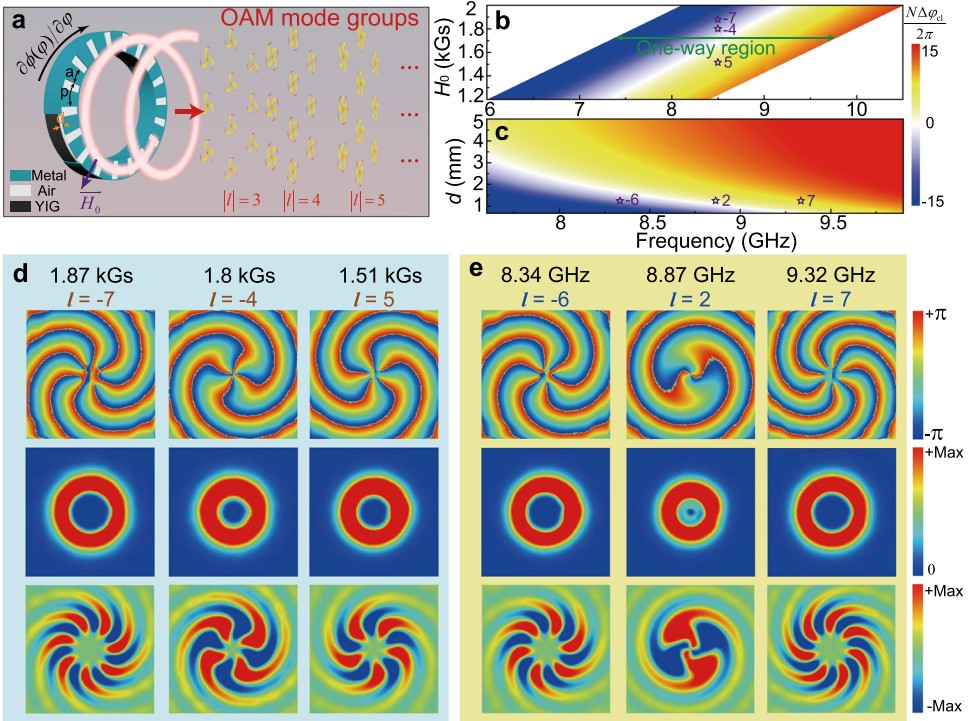

**Fig. 6 | Tunable ring-cavity OAM source based on the unidirectional guided wave-driven metasurface. a** Schematic of a ring-cavity OAM source enabled by the unidirectional guided wave-driven metasurface. Unidirectional phase modulation provided by the metasurface intrinsically breaks the degeneracy of CW and CCW WGMs, leading to multiple OAM radiation at different frequencies. The dependence of USMP dispersion on geometric parameter and external magnetic field ($H_0$) makes the radiated OAM modes tunable. **b, c** Numerical characterization of $N\Delta\varphi_{cl}/2\pi$ from the ring-cavity OAM source with varied external magnetic field ($H_0$) and dielectric depth ($d$) at different frequencies, the integer value of which corresponds to OAM with certain topological charge. **d** Simulated far-field phase (top), intensity (middle) and real part (bottom) profiles of the field component $E_r$ of generated beams tuned through external magnetic field ($H_0$) corresponding to different OAM eigenstates, which are marked in (**b**). **e** The corresponding results of generated OAM beams with different eigenstates for different frequencies, which are marked in (**c**).

## Tunable ring-cavity OAM source

Utilizing 3D meta-cells to construct a ring cavity, we are able to create a tunable source that directly emits OAM beams. OAM beams are considered as potential candidates for encoding information in both classical and quantum optical systems. A conventional system for generating OAM beams is usually based on combining a bulk source with additional phase-front shaping components, and such an approach is not easily scalable and integratable. This fundamental limitation can be overcome by compact and integratable ring-cavity OAM sources[28,30]. However, a ring cavity generally supports degenerate whispering gallery modes (WGMs): clockwise (CW) and counterclockwise (CCW) propagating modes, and these modes are excited simultaneously, thus their carried OAMs cancel each other. It was demonstrated that controllable OAM emission can be generated by implementing a phase-gradient metasurface integrated on a ring cavity[30]. This metasurface breaks the degeneracy of two counter-propagating WGMs so that only one WGM can couple to the free space emission. Here, by using meta-cells to construct a ring cavity, it only supports resonant modes with unidirectionally circulating energy flow, significantly simplifying the design of OAM sources and enabling a large operating bandwidth. Furthermore, the OAM order of the radiated vortex beam can be tuned by varying the frequency or even by tailoring the external magnetic field at a fixed frequency.

We consider a ring cavity that consists of $N$ uniform meta-cells periodically ranged along the azimuthal direction (Fig. 6a). Suppose USMPs in the cavity travel along the CCW direction. The eigen-WGM in the cavity has the feature of azimuthal resonance and, for the $M$th-order mode, the following condition should be satisfied

$$N\Delta\varphi_{cl} = n_{eff}k_0Np = 2\pi M, \tag{12}$$

where $n_{eff}$ is the effective index of the meta-cell and $p$ is the period length. In this cavity, the USMPs are coherently scattered at the locations $\theta_n = 2\pi n/N$, where $n \in [0, N-1]$, resulting in the extracted phases $\varphi_n = 2\pi nM/N$. The extracted phases increase linearly from 0 to $2\pi M$ along the ring perimeter, thereby creating a vortex beam with topological charge $l = M$. As $n_{eff}$ can be continuously tuned over the range $[-1, 1]$ by varying frequency, the OAM order of radiated beam can vary with frequency. More importantly, $n_{eff}$ can be tuned at a fixed frequency through varying radial external magnetic field applied in the YIG material, so we can achieve OAM sources with tunable topological charges. The radial magnetic field can be generated using a pair of coils, as employed in ref. 57. Note that only USMPs with $|n_{eff}| < 1$ can be coupled to the free space emission, which gives a limitation to the range of achievable OAM states.

Let the quantity $l = N\Delta\varphi_{cl}/2\pi$ numerically characterize the phase shift of USMPs traveling over $N$ meta-cells. This quantity corresponds to the OAM order of eigen WGM in a ring cavity that consists of $N$ meta-cells. The quantity $l$ can be calculated using the full-wave FEM simulation for the meta-cell with periodic boundary condition. In the presence of external magnetic field ($H_0$), the tensor components in Eq. (1) become $\mu_1 = 1 - \omega_0\omega_m/(\omega^2 - \omega_0^2)$ and $\mu_2 = \omega_m\omega/(\omega^2 - \omega_0^2)$, where $\omega_0 = 2\pi\gamma H_0$ ($\gamma$ is the gyromagnetic ratio) and $\omega_m$ is the characteristic circular frequency. Figure 6b shows the $l$ value as a function of the external magnetic field and frequency ($f$). Here, we assume $N = 43$, and

the parameters of the meta-cells are $\omega_m = 10\pi \times 10^9$ rad/s ($f_m = 5$ GHz), $p = 6$ mm, $a = 4$ mm, and $d = 1.212$ mm (the dispersion relations of USMPs for type-I and type-II waveguides are displayed in Supplementary Fig. S7), this ring cavity has a radius of 41 mm and a width of 2 mm. As the type-I USMP in the meta-cell only exists in the range from $\omega_0 + 0.5\omega_m$ to $\omega_0 + \omega_m$, where $\omega_0 = 2\pi\gamma H_0$ ($\gamma$ is the gyromagnetic ratio), the frequency region for unidirectional WGMs in the ring cavity varies with $H_0$. In the colormap, some $l$ integers are marked with stars for the frequency $f = 8.5$ GHz, and they are found to be really the OAM orders of WGMs in the corresponding cavity. The phase shift for a single meta-cell is $\Delta\varphi_{cl} = n_{eff}k_0p$, and $n_{eff}$ can be tuned by varying $H_0$ over the range $[-1, 1]$. As a result, the OAM order has at least a tunable range of $[-l_{max}, l_{max}]$, where $l_{max} = \text{int}[Np/\lambda_0]$ with $\lambda_0$ being the designed wavelength. Clearly, the radiated OAM can be adjusted at a fixed frequency through varying external magnetic field $H_0$. In addition, the $n_{eff}$ value varies with frequency, and so does the OAM order $l$ of WGM in the ring cavity. Thus, our ring cavity can output vortex beams with different OAM orders for different frequencies, as indicated in Fig. 6c, where $H_0 = 1785$ Gs. For $d = 1.212$ mm, the OAM order $l$ can vary from $-7$ to 7 over the frequency range [8.28, 9.32] GHz (see Supplementary Fig. S8). Besides, it is also available for our ring source to output vortex beams with the same OAM order over a frequency range by properly adjusting the external magnetic field. Therefore, based on the nonresonant mechanism of the metasurface, our ring OAM source can operate over a wide frequency range.

To verify the controllability of the OAM emission with the external magnetic field $H_0$, as marked in Fig. 6b, we displayed the simulated results of OAM radiation from the ring cavity with three representative magnetic field $H_0 = 1.87$, 1.8, and 1.51 kGs in Fig. 6d at the designed frequency $f = 8.5$ GHz (complete results are displayed in Supplementary Fig. S9). The radiated OAM beams is radially polarized. The electric field $E_r$ has a spiral pattern, and its phase changes by $-7 \times 2\pi$, $-4 \times 2\pi$ and $5 \times 2\pi$ respectively upon one full circle around the center of the vortex, indicating the topological charge $l = -7$, $-4$ and 5, matching our simulation result in Fig. 6b. Moreover, the intensity of the electric fields are spatially distributed in a doughnut shape with a dark core in the center, which are due to the topological phase singularity at the beam axis, where the phase becomes discontinuous. On the other hand, according to Fig. 6c, the OAM charge can also be controlled by varying frequency $f$. To illustrate this, Fig. 6e shows three OAM states of $l = -6$, 2 and 7 at frequencies $f = 8.34$, 8.87, and 9.32 GHz, which agree well with those from our theoretical calculation in Fig. 6c. Furthermore, Fig. 6c indicates that the OAM charge can also be tailored by varying structure parameter $d$ (see Supplementary Fig. S10). Finally, we should indicate that a unidirectional ring cavity does not necessarily require a radial magnetic field, it can also effectively operate with a uniform magnetic field (see Supplementary Fig. S11 for details).

## Discussion

Our USMP-driven metasurfaces, consisting of uniform or nonuniform meta-cells of subwavelength length, provide a highly versatile and compact platform for achieving free-space optical functionalities directly from surface plamonic waves. In the microwave domain, we have experimentally verified the existence of such USMPs in a metal-air-gyromagnetic structure. Different from existing metasurfaces with meta-atoms, which induce abrupt phase shifts in the optical path, the present metasurfaces rely on gradual phase accumulation from the propagation of USMPs to mold optical wavefront at subwavelength scale. Such metasurfaces can provide unparalleled phase controllability to free-space wave propagation based on the unique dispersion of USMP (see Table S1 in Supplementary Information for comparison with prior arts). We have demonstrated the generation of a Bessel beam and wave focusing using these structures at microwave frequencies. In addition, by taking advantage of the unidirectional propagation of USMPs, we used a ring-cavity formed by uniform meta-

cells to generate optical vortices that have a helical wavefront and carry OAM with a designable order. Besides, the dispersion property of USMPs can be flexibly tailored by either geometric parameter or external magnetic field, thus facilitating the realization of dynamic controllability of USMP-driven metasurfaces. We have also demonstrated the tunability of OAM order of radiation with the geometric parameter and magnetic field at a designed frequency. The design strategies presented in the paper allow one to tailor in an almost arbitrary way the phases and amplitudes of an optical wavefront. On the basis of the demonstrated design principle, more complex functionalities can be realized, such as USMP-driven holograms, virtual reality displays, and so forth. Furthermore, as our metasurfaces can control wavefronts at deep subwavelength scales, they are capable to realize desired fine field patterns in near field. Using these structures, we have demonstrated the generation of near-field subwavelength focusing that is difficult to achieve with conventional metasurfaces. Such near-field manipulation can break the optical diffraction limit, thus yielding important applications in optical detection, optical sensing and high-resolution optical imaging. The developed technology can be extended to higher frequency regimes (see Supplementary Fig. S12) and could have major implications for integrated optics[5,43].

## Methods

### Numerical simulations

The theoretical data presented in Fig. 2b and 2f were obtained using the software of MATLAB. All numerical simulations were conducted using the finite element method (FEM), with third-order finite elements and at least 10 mesh steps per wavelength to ensure the accuracy of the calculated results. In both 2D and 3D simulations, unidirectional surface magnetoplasmons (USMPs) were excited by point sources and line currents, respectively, placed in a unidirectional waveguide that was connected to the metasurface system. In the numerical simulations of OAMs generation, we excite the resonant mode of the ring cavity using a radial line current, with a length equal to the width of the YIG, and it is placed at the center of the air layer between the YIG and the upper metal wall. The effects of material absorption of yttrium iron garnet (YIG) were taken into account in the damping coefficient, which was set to $v = 10^{-3}\omega$, where $\omega$ is the operating angular frequency.

To generate free-space wave manipulation with desired phase profile, we first simulated a periodic structure consisting of uniform meta-cells (Fig. 2e) using the FEM. The phase difference between the extracted waves from two neighboring cells was calculated as $\Delta\varphi$. We then systematically varied the geometrical parameters of the meta-cells to obtain phase map (Fig. 2f). By analyzing the desired phase profiles, we obtained the target phase difference distribution $\Delta\phi(x, y)$. Subsequently, we identified the structural parameters $d\{i, j\}$ that correspond to the target phase distribution by searching the phase map. This approach allowed us to accurately determine the detailed structure of the meta-cells required for achieving the desired phase distribution.

To flexibly control the phase and amplitude of extracted wave, L-shaped meta-cells are used to construct metasurfaces with more complex functions. In each meta-cell, the type-I waveguide comprises two sections with different air-layer thicknesses. In the leading section of length $a_i'$, the propagation constant ($k_1$) of USMP can be controlled by adjusting the air-layer thickness ($d_i$), while USMP in the latter section with a relatively thick air layer exhibits a propagation constant approximately equal to that ($k_2$) in the type-II waveguide (of length $b_i$). Thus, the phase change over the meta-cell can be approximated as a function solely dependent on the parameter $d_i$, i.e., $\Delta\varphi_{cl} \approx k_1(d_i)a_i' + k_2(p - a_i') + 2\delta\varphi_{12}$. On the other hand, the extracted amplitude can be readily regulated by manipulating the parameter $a_i$ ($= a_i' + a_i''$). As a result, parameters $d_i$ and $a_i$ exert substantial influence solely over the extracted phase and amplitude, respectively. The determination of structure parameters $d\{i, j\}$ can be accomplished by

searching the database in Fig. 2f. Subsequently, the parameters $a\{i, j\}$ can be determined in a step-by-step manner by satisfying the condition of radiation amplitude of each meta-cell. Consequently, all the parameters $a\{i, j\}$ and $d\{i, j\}$ can be precisely determined. The structural details of the designed USMP-driven metasurfaces were displayed in Supplementary Fig. S13-S15.

## Experimental setup

To effectively couple microwave signals into the unidirectional waveguide with subwavelength sizes, the fabricated waveguide sample is composed of three sections, and the middle section is just the unidirectional waveguide. The first and third sections of the waveguide sample are a rectangular metal waveguide of the sizes 12 × 2 mm that is fully filled with ceramic material, which has a relative permittivity of 90. In the middle section, the rectangular metal waveguide is partly filled by a YIG block of height 10 mm, and there is a gap of 2 mm between the YIG block and the upper metal wall. The used YIG material has a saturation magnetization $4\pi M_s = 750$ Gs, relative permittivity $\varepsilon_m = 13.5$, and resonance linewidth $\Delta H = 5$ Oe. Two SMA ports are embedded in ceramics, and they are perpendicular to the waveguide sample. They are used to respectively feed TE-polarized electromagnetic waves into the waveguide to excite the USMP mode and to receive the signals transmitted through the waveguide. This 3D unidirectional waveguide (with a small width of 3 mm) has the same guiding characteristic as the (2D) metal-air-YIG layered structure.

During measurement, the waveguide sample is vertically placed between two magnetic poles (the diameter of the magnetic poles is 50 mm) of an electromagnet, and the magnetic field between the poles is uniform and adjustable by varying the applied voltage. Thus, the YIG material in the waveguide is magnetized in the lateral direction. Both the two SMA ports are connected to a vector network analyzer (Ceyear 3672E) to test the S-parameters.

## Data availability

Authors can confirm that all relevant data are included in the paper and/or its supplementary information files, and raw data are available upon request from the corresponding author.

## Code availability

The codes used to produce these results are available upon request from the corresponding author.

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

## Acknowledgements
This work was supported by National Natural Science Foundation of China (Nos. 12104401 [S.L.], 62075197 [L.S.]) and Zhejiang Provincial Natural Science Foundation of China (No. Z22F047705 [L.S.]). K.L.T. was supported by the General Secretariat for Research and Technology (GSRT) and the Hellenic Foundation for Research and Innovation (HFRI) under Grant No. 4509. K.L.T.'s part was also carried out within the framework of the National Recovery and Resilience Plan Greece 2.0, funded by the European Union – NextGenerationEU (Implementation body: HFRI) under Grant No. 16909.

## Author contributions
S.L. and L.S. conceived the project, performed the initial analysis and simulations. T.J. fabricated the devices and carried out the measurements. S.L. wrote a first draft of the paper, which was then finalized by input from L.S. and K.L.T. Q.S., H.Z., J.Y. and S.S. contributed to the discussion of the manuscript. L.S. and K.L.T. supervised the work.

## Competing interests
The authors declare no competing interests.
