## [Peer Review File · Nature Communications]

Unidirectional guided-wave-driven metasurfaces for arbitrary wavefront controlREVIEWER COMMENTS:

Reviewer #1 (Remarks to the Author):

This manuscript discussed a unidirectional guided wave driven metasurface formed by nonreciprocal material yttrium-iron-garnet (YIG), metal, and air to achieve different wavefront manipulation. The scattering matrix elements were calculated with simulation, which matched with the experimental measurements. Wave focusing, Bessel beam generation, near-field sub-diffraction focusing, ring cavity orbital angular momentum (OAM) source were simulated as examples of the capabilities of the metasurface. The manuscript is well-written. However, there are several major concerns to be addressed and the work does not match the high-quality standard of Nature Communications. Thus, I cannot support the publication of this manuscript.

1. The authors claimed that there will be benefits using nonreciprocal materials for wave manipulations. However, the examples shown do not provide any evidence of benefits of the proposed metasurface compared with existing designs. The authors need to provide direct comparison between their design and the existing metasurfaces to demonstrate their claim.
2. Nearfield imaging beyond diffraction limit is well known since the high k components of the evanescent signal can be detected. Thus, sub-diffraction focusing is not surprising and is achievable even without the proposed metasurface.
3. Generation of OAM with high quality has been achieved by many different methods including single mode laser ring resonator and supersymmetric waveguides. It is unclear what is the benefit in using the proposed metasurface. Indeed, from the simulation results, the quality of OAMs generated from the proposed metasurface are very poor.
4. The studies of all the metasurface functionalities are based on simulations. There is no experimental demonstration of the proposed functionalities, making it questionable about the challenges on fabrication and characterization of the proposed metasurface. The claim of the advantages of the proposed metasurface comparing with existing ones need to be demonstrated using experiments to provide convincing results.

Reviewer #2 (Remarks to the Author):

Unidirectional propagation of optical waves to prevent backscattering is an interesting phenomenon with rich physics and great potential for applications. In this manuscript, the authors have conducted an insightful examination of two distinct types of unidirectional surface magnetoplasmon modes mainly theoretically and numerically. The study's analysis of the dispersion curves indicates that these modes possess unidirectional propagation characteristics and demonstrate resilience to backscattering during scattering events. Moreover, the authors have innovatively employed metasurfaces to facilitate wavefront shaping of the output light. This design holds significant promise for the development of on-chip meta-devices capable of converting surface waves into free-space light with customized wavefronts. While the design presents an interesting forward-thinking approach, there are certain areas that require further exploration to fully substantiate the feasibility of this concept:

1. The inclusion of comparative results demonstrating subwavelength manipulation, especially in scenarios where the waveguide mode is reciprocal, would be highly beneficial. This would provide a clearer understanding of the unique advantages of the proposed design over conventional methodologies.

2. In the pursuit of achieving more precise subwavelength focusing with reduced spot sizes, it would be advantageous to explore the application of vector beams, such as radially polarized states. This exploration should extend beyond phase shift manipulation to assess whether the meta-device can also alter the polarization states of the output light.

3. A critical aspect for real-world applications is the issue of propagation loss, which currently remains unaddressed in the manuscript. For instance, Figure 2d reveals considerable mode attenuation, as indicated by the S-parameter (~ 10 dB in the passband), which presents a significant challenge for the structural design of the device.

4. The clarity of the experimental structure as depicted in Figure 2c could be enhanced. Its current representation makes it difficult for readers to grasp the intricacies of the experimental setup.

5. The concept of integrating a ring cavity with surface magnetoplasmon modes to create a tunable on-chip source for generating high-quality orbital angular momentum states is quite intriguing. However, the manuscript lacks detailed descriptions of the structural design, including critical parameters like the radius and width of the ring cavity. Furthermore, the methodology to generate a radial magnetic field within the confined space of the ring, as shown in Figure 6(a), is not clearly explained. This omission leaves a gap in understanding the operational principles, particularly regarding the excitation of whispering gallery modes in the cavity.

6. The exploration of applying this design in optical frequencies, especially concerning multilayer photonic structures with a 7nm period, prompts queries about its practicality owing to the intricate fabrication process. Moreover, the diminished thickness of metallic films at this scale brings quantum effects (citations C1-C3) into significant prominence, necessitating their inclusion in the analysis. The present discourse falls short of persuasiveness without addressing these quantum phenomena. Therefore, it might be prudent to reconsider the inclusion of discussions pertaining to the visible spectrum.

[C1] Maniyara, R.A., Rodrigo, D., Yu, R. et al. Tunable plasmons in ultrathin metal films. *Nat. Photonics* 13, 328–333 (2019).

[C2] Zhu, W., Esteban, R., Borisov, A. et al. Quantum mechanical effects in plasmonic structures with subnanometre gaps. *Nat Commun* 7, 11495 (2016).

[C3] Yu, H., Peng, Y., Yang, Y. et al. Plasmon-enhanced light–matter interactions and applications. *NPJ Comput Mater* 5, 45 (2019).

Response Letter to Reviewers

We thank both reviewers for their constructive comments and inputs on our work. The comments have helped us to substantially enrich the content of the revised manuscript. In the following text, the comments from the reviewer are presented in 'black' text, followed by our comprehensive response highlighted in 'blue'. Taking into consideration the reviewer's comment, the revisions incorporated in the revised manuscript and supplementary materials are highlighted in yellow. The modified texts in the revised manuscript are posted here in 'red' font.

Reviewer #1 (Remarks to the Author):

This manuscript discussed a unidirectional guided wave driven metasurface formed by nonreciprocal material yttrium-iron-garnet (YIG), metal, and air to achieve different wavefront manipulation. The scattering matrix elements were calculated with simulation, which matched with the experimental measurements. Wave focusing, Bessel beam generation, near-field sub-diffraction focusing, ring cavity orbital angular momentum (OAM) source were simulated as examples of the capabilities of the metasurface. The manuscript is well-written. However, there are several major concerns to be addressed and the work does not match the high-quality standard of Nature Communications. Thus, I cannot support the publication of this manuscript.

Our reply:

We thank the reviewer for their comments and suggestions. We have carefully considered every question raised by the reviewer, and made corresponding modifications and improvements to the manuscript. We hope that the revised manuscript will well meet the high-quality standard of Nature Communications.

1. The authors claimed that there will be benefits using nonreciprocal materials for wave manipulations. However, the examples shown do not provide any evidence of benefits of the proposed metasurface compared with existing designs. The authors need to provide direct comparison between their design and the existing metasurfaces to demonstrate their claim.

Our reply:

We accept the opinions of the reviewer, and to clarify the benefits of our metasurface, it is necessary to directly compare it with other metasurfaces. Currently, there are many types of proposed metasurfaces, which can be divided into two classes according to the excitation method. The first class is excited by incident waves in free space, including the earliest proposed gradient metasurfaces^{1,2}, while the second class is driven by guided waves. The metasurfaces of the first class can manipulate wavefronts at subwavelength scales and are typically constructed from meta-atoms (subwavelength resonance structures). In contrast to the first class, the second class of metasurfaces can be integrated into chips, creating an access between optical integrated circuits and free-space platforms, which has recently received increasing attention from researchers, and our metasurface just belongs to this class. Recently, there are two representative examples of metasurfaces of the second class, with completely different mechanisms

for extracting and controlling wavefronts. The first example is composed of meta-atoms, and operates by extracting and manipulating wavefronts through the interaction between meta-atoms and the evanescent fields of guided waves, enabling manipulation at subwavelength scales but with relatively complex designs³. The second representative example is the leaky-wave metasurface^{4,5}, which is constructed by directly opening holes on conventional metal waveguides, and it controls phase and amplitude of extracted wavefront through the mixing of local leaky waves staggered by a quarter of guided wavelength within the constituent meta-cell. Its design is simple, but such a metasurface can only manipulate wavefronts at wavelength scales. Our metasurface belongs to this type of leaky-wave metasurface, but its mechanism is completely different from the previous examples. Based on the unique dispersion and robust unidirectionality of USMPs, our metasurface can directly control the phase and amplitude of leaky waves through waveguide design, allowing it to manipulate wavefronts even at deep subwavelength scales, which is superior to all previous designs.

To illustrate the advantages and uniqueness of our metasurface, we have now added the text listed below in the manuscript and Table S1 in the Supplementary Materials. In Table S1, our metasurface is directly compared with representative examples of the first and second categories of metasurfaces.

(On page 2, lines 40-48)

Very recently, leaky-wave metasurfaces (LWM) have been proposed^{39,40}, which are constructed by directly opening holes on conventional metal waveguides. It controls phase and amplitude of extracted wavefront through hybridizing local leaky waves staggered by a quarter of guided wavelength within the constituent meta-cell. However, such metasurfaces can only manipulate wavefronts at wavelength scales. Our metasurfaces represent an innovative class of LWMs distinguished by a distinct mechanism. Leveraging the singular dispersion and robust unidirectionality of USMPs, these metasurfaces possess the capability to flexibly modulate both the phase and amplitude of leaky waves by fine-tuning local waveguide parameters. This capability enables precise manipulation of wavefronts, even at deep subwavelength scales.

(Supplementary Materials, section S14)

Representative Works	Major Principle	Capabilities and Key Advantages	Disadvantages
[20], [21]	Artificially designed meta-atoms impart abrupt phase shift to the incident wave from free space	complete manipulation of amplitude, phase, and polarization of light fields in both near-field and far-field regimes with an ultra-thin planar framework.	On-chip integration becomes difficult when driven by free-space waves. Controlling wavefronts at deep-subwavelength scales remains challenging.
[30]	Dressing metasurfaces onto reciprocal waveguide	Arbitrary far-field beam shaping by extracting light from a waveguide.	Necessitating meta-atoms with complex structures at subwavelength scales. Unable to manipulate wavefronts at deep-subwavelength scales.

Representative Works	Major Principle	Capabilities and Key Advantages	Disadvantages
[39], [40]	Single-layered leaky-wave metasurface supporting quasi-bound states in the continuum	Arbitrary control of far-field beam shaping, including polarization, with four degrees of freedom.	Manipulating wavefronts at subwavelength scales is beyond its capabilities.
This work	Directly extracting EM wave from unidirectional waveguide with unique dispersion	Complete and arbitrary ultra-precise control of near-field and far-field wavefront shapes at deep-subwavelength scales.	Arbitrary control of the polarization of wavefronts remains challenging.

Supplementary Table S1: Comparison of the proposed USMP-driven metasurface with selected prior art.

2. Nearfield imaging beyond diffraction limit is well known since the high k components of the evanescent signal can be detected. Thus, sub-diffraction focusing is not surprising and is achievable even without the proposed metasurface.

Our reply:

In the previous manuscript, we have performed subwavelength focusing in the near field for demonstrating the capability of our metasurface to manipulate wavefronts at deep subwavelength scales. Clearly, our example is too simple to adequately illustrate the benefits of metasurfaces with this capability. Therefore, in the revised manuscript, the example of subwavelength focusing of a single spot was replaced by subwavelength focusing of two spots, where the center-to-center distance of the two spots is only 0.25 wavelength, but the shape of the two spots remains very clear and distinguishable, as shown in Fig. 5 of the revised manuscript. Additionally, we further designed a holographic metasurface for near-field imaging of the Greek letter " ϕ ", which is an array of leaky nonuniform USMP waveguides. This metasurface can form an image in the near field with sizes of the wavelength scale, and both the linewidths of the circular ring and straight line are only $\lambda/5$, as shown in Fig. S6 of the Supplementary Materials. Different from conventional near-field imaging, our holographic metasurface is ultra-thin and is excited by guided waves, making it suitable for integration into chips, thereby demonstrating the uniqueness and advantage of this near-field super-resolution imaging technique.

(On page 11)

Fig. 5 | Demonstration of sub-diffraction dual-spot focusing with a USMP-driven metasurface. **a** Schematic diagram of the USMP-driven metasurface which is composed of 2D L-shaped meta-cells with a deep subwavelength size $p \ll \lambda$. The wavefront extracted from the USMP waveguide can not only contain spatial low-frequency components but also spatial high-frequency components, and the sub-diffraction dual-spot focusing can be realized with the contribution of evanescent fields. **b** Designed amplitude, phase profiles localized, and the simulated amplitude and phase of the extracted wave from each meta-cell. **c** Normalized $|E_y|^2$ at the height $z = 0.1\lambda$, along with the corresponding theoretical results. The two spots are separated by 0.25λ with each hotspot size of 0.127λ . **d** Theoretically computed field pattern $|E_y|^2$ excited by an initial wavefront $E_y = U_0(x)$ of finite size 2λ at $z = 0$. **e-g** Normalized intensity profiles $|E_y|^2$, $|H_x|^2$ and $|H_z|^2$ of the generated sub-diffraction dual-spot focusing beam.

(Supplementary Materials, section S6)

Fig. S6. Demonstration of super-resolution imaging of the Greek letter "phi" with a USMP-

driven metasurface. **a** Schematic diagram of the USMP-driven holographic metasurface for near-field imaging. The meta-cells comprising the metasurface have the deep wavelength size $p = \lambda/20$, where λ is the vacuum wavelength for the design frequency 2.69 GHz. **b** Designed amplitude and phase profiles of the wavefront extracted from the metasurface. **d** Simulated $|E_y|$ pattern on a longitudinal slice of $y = 0$. The dashed line marks the location of the image plane at $z = 0.1\lambda$. **e** Simulated $|E_y|$ pattern on the image plane. **f** Distribution of normalized E_y amplitude along the x axis on the image plane, which indicates that the linewidths of the circular ring and straight line are only 0.2λ .

Accordingly, we have added the following text listed below in the manuscript:

(On page 12, lines 3-4)

The center-to-center distance of the two spots is only 0.25 wavelength, but the shape of the two spots remains very clear and distinguishable.

(On page 12, lines 12-16)

Evidently, given such (deep-subwavelength) manipulation capability, one can design holographic metasurfaces capable of generating images with sub-diffraction resolution in the near field. An exemplary demonstration was provided via numerical simulation, showcasing super-resolution imaging of the Greek letter " ϕ " (see Supplementary Fig. S6).

3. Generation of OAM with high quality has been achieved by many different methods including single mode laser ring resonator and supersymmetric waveguides. It is unclear what is the benefit in using the proposed metasurface. Indeed, from the simulation results, the quality of OAMs generated from the proposed metasurface are very poor.

Our reply:

Ring OAM resonators can be easily integrated into chips, making them significantly advantageous over other methods of generating OAM beams. Compared to previous ring OAM resonators, our ring OAM resonator is completely different in physics, thus possessing unique characteristics and advantages. This ring resonator, constructed with a USMP metasurface, only supports resonant modes with unidirectionally circulating energy flow, greatly simplifying the design of OAM source and enabling a large operating bandwidth. By changing the operation frequency, this OAM source can generate OAM beams of various topological charges. More importantly, at a fixed frequency, OAM beams with various topological charges can also be generated through adjusting the external magnetic field. These characteristics are unattainable with conventional methods of OAMs generation, which make our method of ring-cavity OAM source unique and significant.

To unclearly express the benefits in using the proposed metasurface, we substituted "In such cavity, the inversion symmetry is intrinsically broken, and only one of either CW or CCW WGM is allowed to propagate, which is determined by the magnetization direction in the YIG materials. In contrast to the previous approach using phase-gradient metasurface, only unidirectional power circulation can occur in our cavity, which eliminates the undesired spatial hole-burning effect resulted from the interference pattern of two counter-propagating WGMs. So the targeted WGM in our ring cavity unidirectionally circulates carrying OAM through the azimuthally

continuous phase evolution.” with “Here, by using meta-cells to construct a ring cavity, it only supports resonant modes with unidirectionally circulating energy flow, significantly simplifying the design of OAM sources and enabling a large operating bandwidth. Furthermore, the OAM order of the radiated vortex beam can be tuned by varying the frequency or even by tailoring the external magnetic field at a fixed frequency.” in page 12, lines 30-34 of the revised manuscript.

In our previous manuscript, due to rough design of the ring cavity, the quality of the output OAM beams was somewhat poor. In the revised manuscript, we have redesigned the ring cavity for achieving highly uniform field amplitude of the resonant modes (along the azimuthal direction) within the cavity (by increasing the ratio of a/p) and reducing the influence of the near field around the excitation source (by properly increasing the period p), thus solving the problem of the quality of the output OAM beams, as illustrated in the updated Fig. 6 of the main text.

(On page 19)

Fig. 6 Tunable ring-cavity OAM source based on the unidirectional guided wave-driven metasurface. **a** Schematic of a ring-cavity OAM source enabled by the unidirectional guided wave-driven metasurface. Unidirectional phase modulation provided by the metasurface intrinsically breaks the degeneracy of CW and CCW WGMs, leading to multiple OAM radiation at different frequencies. The dependence of USMP dispersion with geometric parameter and external magnetic field makes the radiated OAM modes tunable. **b, c** Numerical characterization of $N\Delta\varphi_{cl}/2\pi$ from the ring-cavity OAM source with varied external magnetic field (H_0) and dielectric depth (d) at different frequencies, the integer value of which corresponds to OAM with certain topological charge. **d1-f3** Simulated far-field phase, intensity and real part of E_r profiles of generated beams tuned through external magnetic field (H_0) corresponding to different OAM eigenstates, which are marked in (b). **g1-i3** The corresponding results of generated OAM beams with different eigenstates for different frequencies, which are marked in (c).

4. The studies of all the metasurface functionalities are based on simulations. There is no

experimental demonstration of the proposed functionalities, making it questionable about the challenges on fabrication and characterization of the proposed metasurface. The claim of the advantages of the proposed metasurface comparing with existing ones need to be demonstrated using experiments to provide convincing results.

Our reply:

We thank the referee for the valuable raised comment. In our revised work, we first experimentally demonstrate in detail the feasibility and practicality of our designs in the microwave regime, including new experiments we have performed during the revisions stage (please see below), and we then thoroughly, both, theoretically and numerically, validate their novel operation through the design of several microwave meta-devices using metal-air-gyromagnetic unidirectional surface magneto-plasmons, converting unidirectional guided modes into the wavefronts of 3D Bessel beams, focused waves, and controllable vortex beams. We, also, numerically demonstrate *sub-diffraction* focusing, which is currently beyond the capability of conventional metasurfaces. Owing to absence of further experimental equipment in our lab(s), this is the most complete characterization that we can presently undertake: We leave no doubt at all that our devices can practically be built, and we then demonstrate, we hope convincingly, both, theoretically and by corroborating full-wave simulations, several important and novel functionalities, some of which are not currently doable with conventional metasurfaces. We thus feel/hope that the works carries sufficient merit as to be publishable in Nature Communications, given also the fact that we do not expect any particular challenges in the detection and characterization of the radiated waves, other than those customarily being encountered in conventional metasurfaces, too.

In some more detail, we have experimentally confirmed the existence of unidirectional modes by measuring the S-parameters of the waveguide. In the revised manuscript, we have now also added experimental results on modes dispersion, further confirming its unique characteristic, i.e., the dispersion curve covers the entire light cone in air without changing the slope sign (see Fig. S1d); additionally, we have now added experimentally measured propagation lengths (over which the energy flow decays by $1/e$) of the unidirectional mode, which is about 8λ at the center of the unidirectional frequency window (see Fig. S1c). For, both, dispersion and propagation length of the unidirectional mode, the experimental results are found to be in good agreement with theoretical results, indicating the correctness and reliability of the physical model of YIG material in our theoretical study. In our metasurface designs, aside from YIG, the constituent materials are simply air and metal. These structures function as passive devices, allowing for precise numerical simulations. Therefore, the experimental results of the unidirectional waveguide we provide are we feel sufficient to provide reliable physical basis for the feasibility of these metasurface structures.

We have now added the text listed below in the manuscript and elaborated the newly added experiments in detail in the Supplementary Materials, section S1.

(On page 4, lines 37-42)

Within the unidirectional waveguide, the propagation loss of USMP is mainly caused by the loss of the YIG material. At the center (2.7 GHz) of the unidirectional window, the measured propagation length (over which the energy flow decays by $1/e$) of USMP is about $8\lambda_0$ (vacuum wavelength). The unique dispersion of the unidirectional mode was also experimentally demonstrated, and it can be adjusted by varying the external magnetic field (see Supplementary Fig. S1).

(From page 4 line 44 to page 5 line 1)

small propagation loss, and complete phase controllability

(Supplementary Materials, section S1)

Fig. S1. Experimental measurement and numerical simulation of the microwave unidirectional waveguide. **a** The measured S-parameters of the waveguide sample 2. **b** Simulated magnetic-field intensity ($|H|^2$) distributions at 2.69 GHz for forward and reverse transmissions. Top: a schematic diagram of the waveguide sample. **c** Theoretical (solid line), simulated (dashed line), and experimentally measured (star) propagation length as a function of frequency. **d** Simulated (solid lines) and measured (circles) dispersion relations of the unidirectional waveguide for external magnetic fields $H_0 = 300, 400,$ and 500 Oe.

Reviewer #2 (Remarks to the Author):

Unidirectional propagation of optical waves to prevent backscattering is an interesting phenomenon with enrich physics and great potential for applications. In this manuscript, the authors have conducted an insightful examination of two distinct types of unidirectional surface magnetoplasmon modes mainly theoretically and numerically. The study analysis of the dispersion curves indicates that these modes possess unidirectional propagation characteristics and demonstrate resilience to backscattering during scattering events. Moreover, the authors have innovatively employed metasurfaces to facilitate wavefront shaping of the output light. This design holds significant promise for the development of on-chip meta-devices capable of converting surface waves into free-space light with customized wavefronts. While the design presents an interesting forward-thinking approach, there are certain areas that require further exploration to fully substantiate the feasibility of this concept:

Our reply:

We thank the reviewer very much for this positive comments and constructive suggestions on our work. We fully accept these suggestions and have made large revisions to the manuscript according to them. Not only have we updated some contents, but also have added new contents to better illustrate the capabilities and uniqueness of our metasurfaces. The feasibility of these concepts has been substantiated by using specific examples.

1. The inclusion of comparative results demonstrating subwavelength manipulation, especially in scenarios where the waveguide mode is reciprocal, would be highly beneficial. This would provide a clearer understanding of the unique advantages of the proposed design over conventional methodologies.

Our reply:

We agree with the reviewer's comments. To illustrate the unique advantage of our metasurface, it is necessary to make a direct comparison with the metasurfaces based on reciprocal waveguides. Currently, there are two representative examples of such metasurfaces, with completely different mechanisms for extracting and manipulating wavefronts. The first representative example (here referred to as type-I metasurface) is composed of meta-atoms³. It extracts and manipulates wavefronts through the interaction between meta-atoms and the evanescent field of the guided wave, enabling wavefront manipulation at subwavelength scales but with a relatively complex design. The second representative example (type-II metasurface) is leaky-wave metasurface^{4,5}, which is constructed by directly perforating a conventional metal waveguide. It ingeniously utilizes the physical concept of bound states in the continuum spectrum, making a simply structured waveguide has the capabilities to arbitrarily manipulate the phase, amplitude, and even polarization of wavefronts. However, this manipulation (especially for phase) relies closely on the non-local mutual interaction between local leaky waves with different properties, thus limiting the wavefront manipulation only at the wavelength scale.

Our metasurface also belongs to the category of leaky-wave metasurfaces, but its mechanism is completely different from the type-II metasurface. Based on the unique dispersion and robust unidirectionality of USMPs, our metasurface directly controls the phase and amplitude of local leaky waves through waveguide design. Therefore, it can manipulate wavefronts even at deep subwavelength scales, which is superior to the previous two types of metasurfaces. For the type-I metasurface, the size of meta-atoms closely depends on the design wavelength, and it is also necessary to avoid interactions between adjacent meta-atoms, thereby failing to manipulate wavefronts at deep subwavelength scales. Undoubtedly, for the manipulation of wavefronts at subwavelength scales, both the type-I metasurface and our metasurface can achieve high-quality free-space functionalities in the far field. To demonstrate the unique advantage of our metasurface, we have added Table S1 in the Supplementary Materials, where we directly compare our metasurface with type-I and type-II metasurfaces.

Furthermore, to better illustrate the unique advantages of our metasurface, we have updated the example of wavefront manipulation at deep subwavelength scale in the revised manuscript, i.e., the example of single-spot subwavelength focusing has been replaced with dual-spot subwavelength focusing, where the distance between the centers of the two spots is only 0.25 wavelengths, but the dual-spot shape remains highly distinguishable, as shown in revised Fig. 5. In addition, we have added a new example of super-resolution imaging for the Greek letter " ϕ ", and have designed a holographic metasurface, which is an array of leaky nonuniform USMP waveguides. The clear image generated by this metasurface in the near field has sizes of the wavelength scale, and the linewidths of the circular ring and straight line are only $\lambda/5$, as shown in Fig. S6 of the Supplementary Materials. Different to the conventional near-field imaging, our holographic metasurface is ultra-thin and waveguide-excited, thus can be integrated onto chips, highlighting the unique advantage of this super-resolution holographic imaging.

(Supplementary Materials, section S14)

Representative Works	Major Principle	Capabilities and Key Advantages	Disadvantages
[20], [21]	Artificially designed meta-atoms impart abrupt phase shift to the incident wave from free space	complete manipulation of amplitude, phase, and polarization of light fields in both near-field and far-field regimes with an ultra-thin planar framework.	On-chip integration becomes difficult when driven by free-space waves. Controlling wavefronts at deep-subwavelength scales remains challenging.
[30]	Dressing metasurfaces onto reciprocal waveguide	Arbitrary far-field beam shaping by extracting light from a waveguide.	Necessitating meta-atoms with complex structures at subwavelength scales. Unable to manipulate wavefronts at deep-subwavelength scales.

Representative Works	Major Principle	Capabilities and Key Advantages	Disadvantages
[39], [40]	Single-layered leaky-wave metasurface supporting quasi-bound states in the continuum	Arbitrary control of far-field beam shaping, including polarization, with four degrees of freedom.	Manipulating wavefronts at subwavelength scales is beyond its capabilities.
This work	Directly extracting EM wave from unidirectional waveguide with unique dispersion	Complete and arbitrary ultra-precise control of near-field and far-field wavefront shapes at deep-subwavelength scales.	Arbitrary control of the polarization of wavefronts remains challenging.

Supplementary Table S1: Comparison of the proposed USMP-driven metasurface with selected prior art.

(On page 11)

Fig. 5 | Demonstration of sub-diffraction dual-spot focusing with a USMP-driven metasurface. **a** Schematic diagram of the USMP-driven metalens which is composed of 2D L-shaped meta-cells with a deep subwavelength size $p \ll \lambda$. The wavefront extracted from the USMP waveguide can not only contain spatial low-frequency components but also spatial high-frequency components, and the sub-diffraction dual-spot focusing can be realized with the contribution of evanescent fields. **b** Designed amplitude, phase profiles localized, and the simulated amplitude and phase of the extracted wave from each meta-cell. **c** Normalized $|E_y|^2$ at the height $z = 0.1\lambda$, along with the corresponding theoretical results. The two spots are separated by 0.25λ with each hotspot size of 0.127λ . **d** Theoretically computed field pattern $|E_y|^2$ excited by an initial wavefront $E_y = U_0(x)$ of finite size 2λ at $z = 0$. **e-g** Normalized intensity profiles $|E_y|^2$, $|H_x|^2$ and $|H_z|^2$ of the generated sub-diffraction dual-spot focusing beam.

(Supplementary Materials, section S6)

Fig. S6. Demonstration of super-resolution imaging of the Greek letter "φ" with a USMP-driven metasurface. **a** Schematic diagram of the USMP-driven holographic metasurface for near-field imaging. The meta-cells comprising the metasurface have the deep wavelength size $p = \lambda/20$, where λ is the vacuum wavelength for the design frequency 2.69 GHz. **b** Designed amplitude and **c** phase profiles of the wavefront extracted from the metasurface. **d** Simulated $|E_y|$ pattern on a longitudinal slice of $y = 0$. The dashed line marks the location of the image plane at $z = 0.1\lambda$. **e** Simulated $|E_y|$ pattern on the image plane. **f** Distribution of normalized E_y amplitude along the x axis on the image plane, which indicates that the linewidths of the circular ring and straight line are only 0.2λ .

Accordingly, we have added the following text listed below in the manuscript:

(On page 12, lines 3-4)

The center-to-center distance of the two spots is only 0.25 wavelength, but the shape of the two spots remains very clear and distinguishable.

(On page 12, lines 12-16)

Evidently, given such (deep-subwavelength) manipulation capability, one can design holographic metasurfaces capable of generating images with sub-diffraction resolution in the near field. An exemplary demonstration was provided via numerical simulation, showcasing super-resolution imaging of the Greek letter "φ" (see Supplementary Fig. S6).

2. In the pursuit of achieving more precise subwavelength focusing with reduced spot sizes, it would be advantageous to explore the application of vector beams, such as radially polarized states. This exploration should extend beyond phase shift manipulation to assess whether the meta-device can also alter the polarization states of the output light.

Our reply:

At subwavelength scales, our metasurfaces can arbitrarily manipulate the phase and

amplitude of extracted wavefronts, but arbitrary polarization control still seems to be an issue to further study. Perhaps this necessitates a double- or multi-layer structures to address this issue. Nevertheless, for some special vector beams and their focusing, such as radially or azimuthally polarized vector beams and their focusing, it is still easy to design single-layer structures for our metasurfaces to realize them, at least in the microwave regime. To show this, in the microwave regime and by utilizing YIG with residual magnetization, we have designed metasurfaces to generate focused Laguerre-Gaussian beams with radial and azimuthal polarizations. Numerical simulations show that these metasurfaces are indeed capable of achieve these desired free-space functionalities, as illustrated below:

Fig. R1. Focused Laguerre-Gaussian beams generated by USMP-driven metasurfaces. **a** Schematic of designed USMP-driven metasurface for the azimuthal polarization. This metasurface consists of 60 sections along the azimuthal direction, and each section is composed of 30 meta-cells with the same length p and hole size d arranged along the radial direction. Note that for different sections, the hole size d is different. **b** Schematic of designed USMP-driven metasurface for the radial polarization. This metasurface consists of 14 rings, and all rings are composed of meta-cells with the same parameters, but the number of meta-cells decreases as the radius of the ring decreases. **c-f** Normalized intensity distributions of nonzero transverse component $|E_\phi|^2$ or $|E_r|^2$, and the longitudinal component $|E_z|^2$ in the xz plane and the focal (xy) plane for (c,e) azimuthally polarized beam and (b,f) radially polarized beam. The position of the focal plane is indicated by dashed line in c-f. **g,h** Normalized intensities of $|E_\phi|^2$, $|E_r|^2$, and $|E_z|^2$ along the x direction for (g) azimuthally and (h) radially polarized beams, respectively. In the simulation of focused Laguerre-Gaussian beam with azimuthal polarization, the metadvice has an inner radius of $R_1 = 37.2$ mm and an outer radius of $R_2 = 334.8$ mm, and the meta-cells have the parameters $p = 9.92$ mm, $a = p/2 = 4.96$ mm. The excitation source is a circular line

current at the frequency 2.69 GHz, and different values of d for meta-cells at different azimuth angles ensure that the phase extracted from each cell satisfies the required distribution for focusing. For the focused Laguerre-Gaussian beam with radial polarization, the inner and outer radii of the metadvice are $R_1 = 18.9$ mm and $R_2 = 339.2$ mm respectively, and the parameters of the meta-cells are $p = 30$ mm, $a = 26.25$ mm, and $d = 3.91$ mm. The metadvice is simultaneously excited by 14 radial line currents at the frequency 2.69 GHz, and the line current in each ring is positioned at the center of the air layer between the YIG material and the upper metal wall. These line currents within different rings have different phases, ensuring that the phase extracted from each meta-cell meets the required distribution for focusing.

3. A critical aspect for real-world applications is the issue of propagation loss, which currently remains unaddressed in the manuscript. For instance, Figure 2d reveals considerable mode attenuation, as indicated by the S-parameter (~ 10 dB in the passband), which presents a significant challenge for the structural design of the device.

Our reply:

The constituent meta-cell of our metasurfaces is constructed from two types of unidirectional waveguides, which are the metal-air-YIG structure (type-I) and the air-YIG structure (type-II). Obviously, these two types of unidirectional waveguides should have sufficiently low propagation losses, because metasurfaces typically have large dimensions (over 5 wavelengths). In each type of the unidirectional waveguides, the propagation loss of unidirectional mode mainly originates from the absorption of the YIG material. Compared to the type-II waveguide, the absorption loss of the unidirectional mode in the type-I waveguide is relatively large because the portion of modal energy distributed in the air is relatively small.

In order to reduce absorption loss and also to demonstrate the phase controllability of our unidirectional guided-wave, we have re-fabricated two new waveguides using YIG material with reduced resonance linewidth (related to loss coefficient) and lengths of 20 mm and 40 mm. In our updated experiments, the used YIG materials has a saturation magnetization $4\pi M_s = 750$ Gs and resonance linewidth $\Delta H = 5$ Oe. For the parameters of the (type-I) unidirectional waveguide sample, we theoretically calculated the propagation length (over which the energy flow decays by $1/e$) of the unidirectional mode by solving the dispersion equation. We found that the propagation length increases with frequency, and at the center (2.7 GHz) of the unidirectional frequency window, the propagation length is approximately 8λ (λ is the vacuum wavelength), as shown in Fig. S1c of the revised Supplementary Materials. Figure S1c also shows the corresponding results obtained from numerical simulations and experimental measurements, and they are in good agreement with the theoretical results. In the experiments, we measured the S_{21} parameters of two unidirectional waveguides with different lengths (but the other parameters being the same), and the measured results are used to evaluate the propagation length of the unidirectional mode. Here, we should point out that the parameters of the experimental samples have not been optimized for minimizing the propagation loss of unidirectional mode. If the air layer in the waveguide is appropriately thickened or replaced with a dielectric material with a higher permittivity, it is surely possible to further reduce the propagation loss. Additionally, we notice from the literature Ref. 6 that the resonance linewidth of YIG materials may reduce to 0.3 Oe. Therefore, the propagation loss of unidirectional waveguides should not be an obstacle for constructing metasurfaces.

In our experiments of unidirectional waveguides, the measured S_{21} parameter within the unidirectional frequency window is only about -3.7 dB, and the sum of S_{21} and S_{11} is significantly less than 1, but this is not caused by the propagation loss of the unidirectional mode. To effectively couple microwave into the unidirectional waveguide, the experimental waveguide sample actually consists of three sections, and the middle section is a unidirectional waveguide. The first and third sections of the waveguide sample are a rectangular metal waveguide fully filled with ceramic material, which has a relative permittivity of 90. In the middle section, the rectangular metal waveguide is partly filled with YIG material, and there is a gap between the YIG block and the upper metal wall. This 3D unidirectional waveguide has the same guiding characteristic as the (2D) metal-air-YIG layered structure. Because the interface between the ceramic and YIG can also support unidirectional mode, when wave in the unidirectional waveguide (i.e., the middle section) propagates forward to its end, the energy flow is split into two parts, and the large part travels downward along the interface between the ceramic and the YIG, then is trapped by the bottom metal layer, and finally absorbed by the YIG material. Only the smaller part of the energy flow passes through the third section of the waveguide and are experimentally received. So when wave is input in reverse, the value of S_{22} is still low in the unidirectional window, though S_{12} is very low there. Evidently, the input wave excites the unidirectional mode at the interface between the ceramic and YIG. The excited wave propagates downward along the interface, and is trapped at the surface of the bottom metal and finally absorbed by the YIG material. This phenomenon can be clearly observed in our numerical simulations, as illustrated in Fig. S1b. We added the text listed below in the manuscript and elaborated the newly added experiments in detail in the Supplementary Materials, section S1.

(On page 4, lines 37-42)

Within the unidirectional waveguide, the propagation loss of USMP is mainly caused by the loss of the YIG material. At the center (2.7 GHz) of the unidirectional window, the measured propagation length (over which the energy flow decays by $1/e$) of USMP is about $8\lambda_0$ (vacuum wavelength). The unique dispersion of the unidirectional mode was also experimentally demonstrated, and it can be adjusted by varying the external magnetic field (see Supplementary Fig. S1).

(From page 4 line 44 to page 5 line 1)

small propagation loss, and complete phase controllability

(Supplementary Materials, section S1)

Fig. S1. Experimental measurement and numerical simulation of the microwave unidirectional waveguide. **a** The measured S-parameters of the waveguide sample. **b** Simulated magnetic-field intensity ($|H|^2$) distributions at 2.69 GHz for forward and reverse transmissions. Top: a schematic diagram of the waveguide sample. **c** Theoretical (solid line), simulated (dashed line), and experimentally measured (star) propagation length as a function of frequency. **d** Simulated (solid lines) and measured (circles) dispersion relations of the unidirectional waveguide for external magnetic fields $H_0 = 300, 400,$ and 500 Oe.

4. The clarity of the experimental structure as depicted in Figure 2c could be enhanced. Its current representation makes it difficult for readers to grasp the intricacies of the experimental setup.

Our reply:

Thank the reviewer for his reminder. In revising the manuscript, we have provided detailed descriptions of the experimental setup and waveguide samples, namely, we have added the following text in the “Methods” section of the revised manuscript:

(From page 15 line 46 to page16 line 12)

To effectively couple microwave signals into the unidirectional waveguide with subwavelength sizes, the fabricated waveguide sample is composed of three sections, and the middle section is just the unidirectional waveguide. The first and third sections

of the waveguide sample are a rectangular metal waveguide of the sizes 12×2 mm that is fully filled with ceramic material, which has a relative permittivity of 90. In the middle section, the rectangular metal waveguide is partly filled by a YIG block of height 10 mm, and there is a gap of 2 mm between the YIG block and the upper metal wall. The used YIG material has a saturation magnetization $4\pi M_s = 750$ Gs, relative permittivity $\epsilon_m = 13.5$, and resonance linewidth $\Delta H = 5$ Oe. Two SMA ports are embedded in ceramics, and they are perpendicular to the waveguide sample. They are used to respectively feed TE-polarized electromagnetic waves into the waveguide to excite the USMP mode and to receive the signals transmitted through the waveguide. This 3D unidirectional waveguide (with a small width of 3 mm) has the same guiding characteristic as the (2D) metal-air-YIG layered structure.

During measurement, the waveguide sample is vertically placed between two magnetic poles (the diameter of the magnetic poles is 50 mm) of an electromagnet, and the magnetic field between the poles is uniform and adjustable by varying the applied voltage. Thus, the YIG material in the waveguide is magnetized in the lateral direction. Both the two SMA ports are connected to a vector network analyzer (Ceyear 3672E) to test the S-parameters.

In the revised manuscript, we also updated Fig.2c to clarify the experimental structure more clearly.

(On page 5)

Fig. 2 | Phase controllability of the unidirectional guided wave. **a** Schematic diagram of waveguides supporting type-I and type-II USMPs at microwave frequencies, and **b** the dispersion relations of USMPs for various thicknesses of the dielectric layer. The shaded rectangular area indicates the unidirectional frequency window for the waveguide, and the other shaded areas indicate the zones of bulk modes in the gyromagnetic materials ($\epsilon_m = 15$). Inset: simulated electric-field intensity distribution excited by a line current source placed in the middle of waveguide structure with $d = 0.06\lambda_m$ and frequency $\omega = 0.75\omega_m$. **c** Left: photograph of the fabricated waveguide sample. Right: photograph of the measurement configuration. **d** Simulated (solid) and measured (dashed) S_{21} and S_{12} parameters of the waveguide sample

displayed in (c). **e** Schematic picture of the periodically uniform metasurface formed by alternating type-I and type-II waveguides shown in (a). The period length, duty cycle, and dielectric-layer thickness are denoted by p , a/p , and d , respectively. The phases of extracted waves from two neighboring meta-cells are φ_i and φ_{i+1} with a difference $\Delta\varphi$. **f** Pseudocolor map of simulated $n_{\text{eff}} = \Delta\varphi_{\text{cl}}/(k_0p)$ for the structure of (e) in a parameter space spanned by dielectric thickness (d) and duty cycle (a/p) at 2.69 GHz. The three black lines indicate n_{eff} covering $[-1, 1]$ range. The black stars indicate the theoretical results obtained from Eq. (4).

5. The concept of integrating a ring cavity with surface magnetoplasmon modes to create a tunable on-chip source for generating high-quality orbital angular momentum states is quite intriguing. However, the manuscript lacks detailed descriptions of the structural design, including critical parameters like the radius and width of the ring cavity. Furthermore, the methodology to generate a radial magnetic field within the confined space of the ring, as shown in Figure 6(a), is not clearly explained. This omission leaves a gap in understanding the operational principles, particularly regarding the excitation of whispering gallery modes in the cavity.

Our reply:

We are sorry for that did not provide complete information about the ring cavity in the previous manuscript. In the revised manuscript and Supplementary Materials, all missing information has been added.

In the revised manuscript (page 13, line 23), we have added the following sentence: “this ring cavity has a radius of 41 mm and a width of 2 mm.” Here, the width of the ring cavity only refers to the radial width of YIG. In our numerical simulations, the thickness of the metal wall is considered to be 1 mm. If the thickness of the metal wall is included, the width of the ring cavity becomes 4 mm.

To indicate the excitation method of the ring cavity, we have added the following sentences in the “Methods” section of revised manuscript (page 15, lines 16-19): “In the numerical simulations of OAMs generation, we excite the resonant mode of the ring cavity using a radial line current, with a length equal to the width of the YIG, and it is placed at the center of the air layer between the YIG and the upper metal wall.” This excitation method corresponds to using a probe to excite waves at microwave frequencies in experiment.

In the microwave regime, where the dimensions of the ring cavity are generally large, radial magnetic field can be generated using a pair of coils, as employed in Ref. 7. The currents in these two coils are equal in magnitude but opposite in direction. On the mid-plane of the gap between the two coils, the magnetic fields produced by them have opposite vertical components and identical radial components. Therefore, the magnetic field on the mid-plane completely points in the radial direction, and the magnitude of this radial magnetic field can be adjusted by varying the currents. It should be noted that the skin depth of the field of the unidirectional mode in the YIG is very small, and the modal properties primarily depend on the external magnetic field within the skin

depth. Thus, the surface of the YIG should be positioned on the mid-plane between the two coils. In the revised manuscript (page 12, lines 47-48), we have added the following sentence: “The radial magnetic field can be generated using a pair of coils, as employed in Ref. 57.”

In fact, a unidirectional ring cavity does not necessarily require a radial magnetic field, it can also operate within a uniform magnetic field. Suppose to construct such a ring cavity using a unidirectional waveguide, which is a radially layered metal-air-YIG-metal structure. This unidirectional waveguide is terminated by a pair of metal slabs in the axial direction (z direction) of the ring cavity. Under a uniform magnetic field in the z direction, this structure can support resonant modes with unidirectionally circulating energy flows. If periodic holes are further made on the upper metal wall of the ring cavity, this structure is capable to generate OAM beams, possessing the same functionalities of the former unidirectional ring cavity. To demonstrate this, we have added a new example of ring OAM resonator in the section S11 of Supplementary Materials. Our numerical simulations show that the designed structure can generate OAM beams under a uniform external magnetic field. Clearly, this design concept can be extended to the terahertz regime, enabling the construction of unidirectional ring cavities that can generate terahertz OAM beams. Accordingly, we have added the following text listed below in the manuscript:

(On page 14, lines 26-28)

Finally, we should indicate that a unidirectional ring cavity does not necessarily require a radial magnetic field, it can also effectively operate with a uniform magnetic field (see Supplementary Fig. S11 for details).

(Supplementary Materials, section S11)

Fig. S11. Tunable ring-cavity OAM source based on the USMP-driven metasurface operating with a uniform external magnetic field. a Schematic of a ring-cavity OAM source. **b1-d3** Profiles

of simulated far-field phase, intensity and real part of E_z of generated OAM beams at different frequencies. The geometric parameters of the ring cavity are as follows: the ring radius $R = 41$ mm, $p = 6$ mm, $a = 5$ mm, $d = 1$ mm. Here, $H_0 = 1785$ Gs, $p = 6$ mm, $a = 5$ mm, $d = 1$ mm, and the diameter of the YIG ring is 82 mm. The external magnetic field points in the z-axis direction, with the amplitude $H_0 = 1785$ Gs.

6. The exploration of applying this design in optical frequencies, especially concerning multilayer photonic structures with a 7nm period, prompts queries about its practicality owing to the intricate fabrication process. Moreover, the diminished thickness of metallic films at this scale brings quantum effects (citations C1-C3) into significant prominence, necessitating their inclusion in the analysis. The present discourse falls short of persuasiveness without addressing these quantum phenomena. Therefore, it might be prudent to reconsider the inclusion of discussions pertaining to the visible spectrum.

[C1] Maniyara, R.A., Rodrigo, D., Yu, R. et al. Tunable plasmons in ultrathin metal films. *Nat. Photonics* 13, 328–333 (2019).

[C2] Zhu, W., Esteban, R., Borisov, A. et al. Quantum mechanical effects in plasmonic structures with subnanometre gaps. *Nat Commun* 7, 11495 (2016).

[C3] Yu, H., Peng, Y., Yang, Y. et al. Plasmon-enhanced light–matter interactions and applications. *NPJ Comput Mater* 5, 45 (2019).

Our reply:

Thank the reviewer very much for his comments, and we fully accept them. At the present stage, it lacks scientific foundation to discuss USMPs and related metasurfaces in the visible regime. Therefore, discussion and related content regarding visible USMP have been removed from the revised manuscript and Supplementary Materials.

The discussions regarding the extension to the terahertz regime is retained, but we have updated the previous model of terahertz unidirectional waveguide in the revised Supplementary Materials. The new model is an (electric) opaque medium-silicon-InSb layered structure. We find that USMP in this structure exhibits unique dispersion at terahertz frequencies, i.e., its dispersion curve covers the entire light cone in air (see Fig. S12), like USMPs in the microwave regime. Such unidirectional waveguides are suitable for constructing terahertz USMP-driven metasurfaces.

(Supplementary Materials, section S12)

Fig. S12. Robust USMPs with complete phase controllability in terahertz regime. a Schematic of the plasmonic structure supporting robust USMP at terahertz frequencies. **b, c** Dispersion curves of USMPs in the upper bandgap of the magnetized semiconductor for different d_r and B_0 values. The shaded areas represent the zones of bulk modes in the magnetized semiconductor, and the dashed lines represent light lines in free space. The external magnetic field in **b** is $B_0 = 0.5$ T, and the thickness of the dielectric layer in **c** is $d_r = 0.054\lambda_p$ (λ_p is the vacuum wavelength for ω_p). The other parameters of the guiding system are $\epsilon_r = 11.68$, $\epsilon_m = -7$, and $d_m = 0.1\lambda_p$.

References

1. Yu, N. F. et al. Light propagation with phase discontinuities: generalized laws of reflection and refraction. *Science* 334, 333–337 (2011).
2. Sun, S. L. et al. Gradient-index meta-surfaces as a bridge linking propagating waves and surface waves. *Nat. Mater.* 11, 426–431 (2012).
3. Guo, X. X., Ding, Y. M., Chen, X., Duan, Y. & Ni, X. J. Molding free-space light with guided wave-driven metasurfaces. *Sci. Adv.* 6, eabb4142 (2020).
4. Xu, G. Y. et al. Arbitrary aperture synthesis with nonlocal leaky-wave metasurface antennas. *Nat. Commun.* 14, 4380 (2023).
5. Huang, H. et al. Leaky-wave metasurfaces for integrated photonics. *Nat. Nanotechnol.* 18, 580–588 (2023).
6. Pozar, D. M. *Microwave Engineering* (John Wiley and Sons Ltd, 2011).
7. Boxman, R. L., Gerby, E. & Goldsmith, S. Behavior of a high current vacuum arc between hollow cylindrical electrodes in a radial magnetic field. *IEEE T. Plasma Sci.* 8, 308-313 (1980).

REVIEWERS' COMMENTS

Reviewer #1 (Remarks to the Author):

The authors didn't really address my major concerns. The proposed design doesn't provide significant advantages comparing with existing metasurfaces. Subwavelength focusing in the near-field is not surprising. Systems like NSOM with similar functionality has been in use for decades. The OAM generation is not new, either. Same design has been published 8 years ago [Science, 353, 464-467 (2016)]. Not even saying that all the major results are simply simulations, and the added experiment was only for unit cell characterization and cannot be used to demonstrate the claimed functionalities. With all these in mind, I don't think this manuscript will fit in high-quality journals like Nature Communications. Thus, I cannot support the publication of this manuscript even with the revision.

Reviewer #2 (Remarks to the Author):

I have now reviewed the revised manuscript titled "Unidirectional guided-wave-driven metasurfaces for arbitrary wavefront control" for the second time. The authors have significantly improved their work based on the feedback provided during the initial review process. By analyzing the reciprocal metasurface reported in the previous works, the authors have explained the advantages of the proposed design based on unidirectional SPP. Furthermore, the addition of near-field super-resolution imaging by holographic metasurface effectively demonstrates the capability to manipulate light at deep subwavelength scales. For the problem of low transmission power through YIG waveguide, the authors have described the experimental setup, and clearly explained the reason. Additionally, the inclusion of structural parameters in the Supplementary Materials enhances the reproducibility and understanding of the proposed design.

While I still maintain reservations regarding the feasibility of the designed metasurface for applications in optical frequencies (no technology is perfect), I appreciate the authors' acknowledgment of this limitation by removing the discussion on optical frequencies in the revised manuscript. The enhanced focus on the feasibility of its implementation in the THz domain represents a more realistic and promising direction for future research. Although the current version does not completely resolve the challenges associated with optical frequencies, the authors' efforts to address this issue and emphasize the potential in the THz range are commendable. Therefore, I recommend accepting this manuscript for publication in its current form.

Response Letter to Reviewers

To reviewer 1:

We are indeed sorry that we are, unfortunately, compelled to respectfully but firmly disagree with the further comment received by the reviewer, for the concise, clear reasons outlined below:

Comment 1: The authors didn't really address my major concerns. The proposed design doesn't provide significant advantages comparing with existing metasurfaces. Subwavelength focusing in the near-field is not surprising. Systems like NSOM with similar functionality has been in use for decades.

Our reply:

In the abstract and introduction of the last revised manuscript, we have clearly stated that our proposed metasurface has a significant advantage over existing ones, namely, its ability to manipulate wavefronts at deep subwavelength scales. To further clarify this point, we have added Table S1 in the revised Supplementary Materials, where our metasurfaces are directly compared with representative existing metasurfaces. Based on this unique advantage, our metasurfaces can not only realize various free-space functionalities in the far field, but also achieve numerous near-field functionalities. In the original manuscript, the subwavelength focusing example was mainly used to illustrate the unique capability of our metasurfaces. Unlike the NSOM technique, the significance of such example for metasurfaces is not limited to mere focusing. In the revised manuscript, to better illustrate this unique capability, we have replaced the original single-spot focusing example with a dual-spot focusing example and added another example of super-resolution holographic imaging, the details of which are presented in the revised Supplementary Materials. Evidently, the reviewer did not carefully consider our revisions and did not provide objective evaluations for them.

Comment 2: The OAM generation is not new, either. Same design has been published 8 years ago [Science, 353, 464-467 (2016)].

Our reply:

Our research work on the ring-cavity OAM source is very different from that mentioned by the reviewer. Firstly, our OAM source is based on new physical mechanism, which is entirely different from that of previous OAM source. Secondly, compared to the previous OAM source, our OAM source has significant advantages. The key technology of the ring-cavity OAM sources is to realize resonant modes of unidirectionally circulating energy flow. In the previous work, unidirectional resonant mode was achieved based on an exceptional point of non-Hermitian Hamiltonian system (with parity-time symmetry) in optics. In contrast, we directly utilize

unidirectional mode of surface magnetoplasmons, which can possess broadband characteristic. Such unidirectional mode can be sustained by the platform of magneto-optical material with bandgap of nontrivial topological property, and it can exist in the entire bandgap. Moreover, the central frequency and width of the bandgap of the magneto-optical material can be tuned by the applied external magnetic field. Based on the unique mechanism of our OAM source, it can have the following significant advantages: 1) The topological charge (including its sign) of the output OMA beam can be tuned by varying the frequency; 2) At a fixed frequency, the topological charge (including its sign) of the output OMA beam can be adjusted through the external magnetic field. Obviously, these merits of our OAM source will be beneficial for its practical applications.

Comment 3: Not even saying that all the major results are simply simulations, and the added experiment was only for unit cell characterization and cannot be used to demonstrate the claimed functionalities. With all these in mind, I don't think this manuscript will fit in high-quality journals like Nature Communications. Thus, I cannot support the publication of this manuscript even with the revision.

Our reply:

The capabilities of our metasurfaces to arbitrarily manipulate the phase and amplitude of wavefronts originate from the unidirectional propagation and unique dispersion of surface magnetoplasmons. In our manuscript, we have provided very clear physical pictures and detailed mathematical descriptions of the principles and methods for manipulating the wavefront phase and amplitude with such metasurfaces. Therefore, it can be judged physically that our proposed metasurfaces are practically feasible as long as there truly exists such unidirectional mode with unique dispersion. Our numerical simulations further confirm this point. In the microwave regime, we have theoretically and experimentally demonstrated true unidirectional mode with unique dispersion using YIG material, and relevant contents (including the propagation length of the unidirectional mode) have been presented in the revised manuscript and Supplementary Materials. The reason why we did not further conduct experimental research on the metasurfaces is only due to the current lack of objective experimental conditions. We feel that the reviewers' comments mentioned above didn't be made from a scientific perspective but rather seems to be based on prejudice.

To reviewer 2:

We thank the reviewer very much for their careful reading of the revised manuscript and Supplementary Materials. All comments from the reviewer are positive, and we are indeed gratified with and thankful for that.